# Annual carbon dioxide flux over seasonal sea ice in the Canadian Arctic

Brian J. Butterworth<sup>1,2</sup>, Brent G. T. Else<sup>3</sup>, Kristina A. Brown<sup>4</sup>, Christopher J. Mundy<sup>4</sup>, William J. Williams<sup>5</sup>, Lina M. Rotermund<sup>6</sup>, Gijs de Boer<sup>7</sup>

- <sup>1</sup> Cooperative Institute for Research in Environmental Sciences, University of Colorado, Boulder, CO, USA
- <sup>2</sup>NOÂA Physical Sciences Laboratory, Boulder, CO, USA
- <sup>3</sup> Department of Geography, University of Calgary, Calgary, AB, Canada
- <sup>4</sup> Centre for Earth Observation Science, University of Manitoba, Winnipeg, MB, Canada
- <sup>5</sup> Fisheries and Oceans Canada, Institute of Ocean Sciences, Sidney, BC, Canada
- <sup>6</sup> Department of Oceanography, Dalhousie University, Halifax, NS, Canada
- <sup>7</sup> Environmental Science and Technologies, Brookhaven National Laboratory, Upton, NY, USA
- 15 Correspondence to: Brian J. Butterworth (brian.butterworth@colorado.edu)

Abstract. Continuous measurements of carbon dioxide (CO2) flux were collected from a 10-m eddy covariance tower in a coastal-marine environment in the Canadian Arctic Archipelago over the course of a 17-month period. The extended length of data collection resulted in a unique dataset that includes measurements from two spring melt and summer seasons and one autumn freeze-up. These field observations were used to verify findings from previous theoretical and laboratory experiments investigating air-sea gas exchange in connection with sea ice. The results corroborated previous findings showing that thick ice cover under winter conditions acts as a barrier to gas exchange. In the spring, CO<sub>2</sub> fluxes were downward (uptake) in both the presence of melt ponds and during ice break-up. However, diurnal cycles were present throughout the early spring melt period, corresponding to the opposing influences of freezing and melting at the ice surface. Fluxes measured during melt periods confirmed previous laboratory tank measurements that showed a gas transfer coefficient of melting ice of 0.4 mol m<sup>-2</sup> d-1 atm-1. Open water CO<sub>2</sub> fluxes showed outgassing in early summer and uptake in mid-to-late summer, tied closely to trends in surface water temperature and its effect on the partial pressure of CO2 in the water. The autumn period of the field campaign represents the first eddy covariance CO2 fluxes measured over naturally forming sea ice. Our measurements showed mean upward fluxes (outgassing) of  $1.1 \pm 1.5$  mmol m<sup>-2</sup> d<sup>-1</sup> associated with the freezing of ice – the same order of magnitude found by previous laboratory tank experiments. However, peak flux periods during ice formation had measured fluxes that were a factor of 3 higher than the tank experiments, suggesting the importance of natural conditions (e.g., wind) on air-ice gas exchange. Conducting an Arctic-wide extrapolation we estimate CO2 outgassing from the freezing period to be a counterbalance equivalent to 5 to 15% of the magnitude of the estimated Arctic CO<sub>2</sub> sink. Overall, there was no evidence of dramatically enhanced gas exchange in marginal ice conditions as proposed by previous studies. Although the different seasons

Deleted: full sea

Deleted: winter

showed active  $CO_2$  exchange, there was a balance between upward and downward fluxes at this specific location, resulting in a small net  $CO_2$  uptake over the annual cycle of -0.3 g-C m<sup>-2</sup>.

#### 1 Introduction

- Polar seas are thought to play an important role as an oceanic carbon sink. Bates & Mathis (2009) estimated that Arctic seas absorb 66–199 Tg-C yr<sup>-1</sup>, or about 5–14% of the global ocean sink, despite only accounting for 3% of the global ocean surface area. More recent studies have produced estimates at the higher end of this range (e.g., 180 Tg-C yr<sup>-1</sup> estimated by Yasunaka et al. [2016], 153 Tg-C yr<sup>-1</sup> estimated by Manizza et al. [2019]). The sea ice zone of the Southern Ocean also absorbs a significant amount of CO<sub>2</sub>, with contemporary estimates suggesting it may be responsible for approximately 4% of the global CO<sub>2</sub> sink (Takahashi et al., 2009), essentially matching its relative surface area (~3%).
  - Unfortunately, most estimates of carbon uptake by polar seas do not rigorously account for CO<sub>2</sub> exchange that might occur between the atmosphere and the sea ice itself, or through fractured icescapes that commonly occur in the fall freeze-up and spring break-up seasons. Some efforts to include these exchanges and the seasons in which they occur have been made. For example, Else et al. (2013) attempted to include CO<sub>2</sub> exchange through winter flaw leads and polynyas in the western Canadian Arctic, and calculated an enhancement to the annual budget of approximately 50%. At a much larger scale, Delille et al. (2014) estimated that direct CO<sub>2</sub> uptake by sea ice might more than double the carbon sink budget of the Southern Ocean's sea ice zone. More recently, Prytherch and Yelland (2021) used eddy covariance measurements near a central Arctic Ocean lead to develop a lead-specific gas transfer velocity parameterization during the summer to fall transition period. Additionally, summertime ship-based Arctic eddy covariance measurements by Dong et al. (2021) showed that surface stratification of fresher, cooler melt water resulted in lower surface pCO<sub>2w</sub> compared to 6-m deep pCO<sub>2w</sub>, with resulting implications for estimating carbon budgets of polar oceans. While these efforts identify specific processes, more measurements are required to quantify additional gas exchange processes over the annual cycle, as well as validate previous findings.
- Direct air-ice CO<sub>2</sub> exchange occurs because sea ice contains gases, both in the dissolved phase (in brine inclusions) and in the gaseous phase (in bubbles), and direct CO<sub>2</sub> exchange can also occur with the precipitation and dissolution of CaCO<sub>3</sub> within the ice brine network (Geilfus et al., 2012). Most observations of CO<sub>2</sub> transfer between sea ice and the atmosphere have been made using the flux chamber method, where a small enclosure (typically ~0.1 m<sup>2</sup>) is placed on the ice and the change in CO<sub>2</sub> concentration over time is recorded. Such studies have found that during initial ice formation, sea ice releases CO<sub>2</sub> at a rate of approximately +1 to +4 mmol CO<sub>2</sub> m<sup>-2</sup> d<sup>-1</sup> (Nomura et al., 2006, 2018). This exchange decreases and eventually ceases as the ice cools and brine connectivity is restricted due to freezing, although the duration of outgassing is not well known (Nomura et al., 2018). CO<sub>2</sub> exchange resumes again in spring with uptake from the atmosphere at rates on the order of -1 to -5 mmol CO<sub>2</sub> m<sup>-2</sup> d<sup>-1</sup> (e.g., Nomura et al., 2013; Geilfus et al., 2015) as warming restores brine channel connectivity, and melt dilution

**Deleted:** However, both of these examples derived their estimates based on very few actual measurements of CO<sub>2</sub> exchange in these challenging environments.

Formatted: Font color: Accent 1

Deleted: salts

and primary production lower brine pCO2. Melt ponds also form on the surface of the ice providing another surface for CO2 uptake (Geilfus et al., 2015).

While insights from flux chamber studies form much of our understanding of direct air-ice exchange, the technique is limited by its restricted spatial scale, inability to produce continuous measurements, and inherent modification of the measured environment (Miller et al., 2015). Eddy covariance – which determines CO<sub>2</sub> exchange via high-frequency measurements of turbulence and CO<sub>2</sub> mixing ratio using instruments located above the surface – does not suffer from these limitations but has rarely been deployed successfully over sea ice. Past attempts have suffered from instrument biases related to the cold marine environment, producing flux estimates that have at times exceeded flux chamber estimates by several orders of magnitude (e.g., Papakyriakou and Miller, 2011; Sievers et al., 2015). Fortunately, recent advancements in eddy covariance system design have reconciled flux magnitudes measured by the two techniques over sea ice (Butterworth and Else, 2018), which should allow for the determination of air-ice CO<sub>2</sub> flux at spatial and temporal scales that are more useful for the development of annual budgets, and without isolating the ice from the atmospheric conditions (particularly wind) that may actually drive fluxes.

In marginal ice environments (mixtures of sea ice and open water), there has been debate about the rate at which CO2 is transferred through the open water portions of the icescape. The primary determinant of gas transfer across an air-water interface is near-surface turbulence, and in the open ocean this turbulence is driven primarily by wind-generated waves (Wanninkhof et al., 2009). In a marginal ice environment, waterside turbulence is expected to be strongly influenced by sea ice through a complex combination of wave attenuation, drag induced by drifting floes, and buoyancy effects that can either drive convection during ice formation or be limited by stratification during ice melt (Loose et al., 2014). Laboratory (Lovely et al., 2015) and tracer-based field studies (Loose et al., 2017) have provided some evidence that factors other than wind speed can contribute to increasing gas transfer velocity in marginal ice environments, and lead to enhanced gas fluxes in some situations. An attempt to study these processes using eddy covariance found CO2 flux enhancement of 1-2 orders of magnitude in a winter flaw-lead polynya (Else et al., 2011), although it now seems likely that this study was affected by the previously mentioned sensor bias problems. Subsequent eddy covariance studies, which used appropriate techniques to eliminate sensor bias, found no enhancement of gas exchange in the presence of sea ice (Butterworth and Miller, 2016a; Prytherch et al., 2017; Prytherch and Yelland, 2021). Recent results from the year-long MOSAiC drift suggest that in marginal ice environments, sea ice largely inhibits gas exchange due to fetch limitation in the open water portions of the icescape (Loose et al., 2024). Resolving this debate is a major impetus for future field studies, particularly given the importance of properly capturing these processes in models of future air-sea CO<sub>2</sub> exchange and ocean acidification rates in the Arctic (Steiner et al., 2013).

In this paper, we provide results from an eddy covariance tower deployed on a small island in the Canadian Arctic Archipelago (Butterworth and Else, 2018). The tower is uniquely positioned to measure fluxes over a surface that experiences complete ice cover in winter, melt pond and marginal ice conditions in spring, complete open water through the summer, and then marginal

ice conditions and eventually full ice cover in fall. Here, we present continuous measurements from the station covering most of this annual cycle with objectives to:

- Present the first annual budget of CO<sub>2</sub> exchange over a sea ice region constructed entirely from eddy covariance
  measurements and to describe the processes that drive seasonal variations in measured exchange.
- 2. Quantify the relative contributions of direct exchange with ice, and exchange in marginal ice conditions, to the overall CO<sub>2</sub> flux budget.
- Look for evidence of meteorological controls on air-ice CO<sub>2</sub> exchange that may not have been captured by past chamber measurements, and for evidence of enhanced CO<sub>2</sub> exchange in marginal ice conditions during freezeup.

#### 2 Methods

#### 2.1 Site description

The eddy covariance tower was installed in April 2017 on the northwest side of Qikirtaarjuk Island in Dease Strait, roughly

35 km west of Cambridge Bay, Nunavut (Fig. 1). The island is small at roughly 500 x 200 m in horizontal extent, with a
maximum elevation of 7 m. It is a rocky island, with essentially no vegetative cover. It is the southernmost island in the
Finlayson Island chain that stretches across the strait. Flux measurements from the tower experience unimpeded fetch from
the east to west. The nearest land to the tower is Unihitak Island, which is 3.5 km to the north, well outside the flux footprint
of the 10-m tower, ensuring that fluxes were entirely from the sea surface during conditions with favorable wind directions.

Southerly winds were discarded during analysis because they pass over the island, as well as through the tower structure.

Figure 1. Map (a) showing the location of Qikirtaarjuk Island, 35 km west of Cambridge Bay, Nunavut. Satellite image (b) of Qikirtaarjuk Island (28 June 2017), showing polynya development in the tidal straits. Circular inset shows the shows the shape of Qikirtaarjuk Island, with the red dot indicating the location of the flux tower. Landsat-8 image courtesy of the U.S. Geological Survey. This figure is reprinted from Butterworth and Else 2018.

The annual sea state in front of the tower (Fig. 2) changes through the year from full sea ice cover in winter (Dec – May), to melt ponds and ice break up in the spring (June – July), full open water in summer (Aug – Sep), and a freeze-up period in the fall (Oct – Nov). Such a seasonal cycle is typical for most of the southern waterways of the Canadian Arctic Archipelago.

Figure 2. Photographs showing the seasonal sea surface conditions in front of the flux tower where (a) shows full ice coverage on 5 Nov 2017 (b) shows melt ponds on 23 June 2017, (c) shows open water on 6 Aug 2017, and (d) shows freezing on 22 Oct 2017. Images a, c, and d taken using GOPRO Hero4 camera mounted at the top of the tower. Image b taken with handheld camera.

# 2.2 Instrument setup

A detailed description of the instrument setup is provided in Butterworth and Else (2018). The main components of the system are the 3-dimensional ultrasonic anemometer (CSAT3; Campbell Scientific) used to measure wind speed in three dimensions and a closed-path infrared gas analyzer (LI-7200; Li-Cor) for measuring CO<sub>2</sub> mixing ratio. Both measurements were made at a sampling frequency of 10 Hz. Unlike previous Arctic eddy covariance systems, this system dried the sample airstream using a moisture exchanger (Nafion; PermaPure) prior to running it through the gas analyzer in order to reduce CO<sub>2</sub> measurement errors associated with water vapor (Miller et al., 2010; Blomquist et al., 2014; Landwehr et al., 2014; Butterworth and Miller, 2016b). For this study, data reported cover the period of May 2017 to September 2018. Data collection was interrupted between January – May 2018 due to failure of the power system during the dark polar winter.

In addition to tower measurements there were water temperature and conductivity measurements made from three different depths (13, 22, and 39 m) on a mooring 1 km north of the tower (68.9930° N, -105.8437° W). The conductivity measurement was used to calculate salinity. In addition, sea surface temperature estimates were obtained from the NOAA ESRL Advanced Very High Resolution Radiometer (AVHRR).

To characterize seasonal sea ice, several methods were deployed. First, images of the sea surface were captured by two cameras (a GoPro Hero4 and a Campbell Scientific CC5MPX) mounted at the top of the tower. Sea ice concentration (SIC) was manually estimated for each image based on a visual assessment of the ice in the immediate foreground (~200 meters) of the tower. During a 2-month stretch from late May to mid-July 2017 both tower-mounted cameras failed and the SIC variable was estimated using a variety of remotely sensed (Landsat-8 and MODIS) and in situ images. These additional in situ images were obtained from a motion-sensor trail camera installed at the base of the tower and from 4 helicopter trips to the island. The manually-derived SIC product showed good agreement with the AMSR-2 passive microwave SIC (daily, 3.125 km) from the University of Bremen (Fig. 3a,b; Spreen et al., 2008), but was deemed preferable due to its representation of the area immediately in front of the tower (i.e., the flux footprint), rather than the larger marine region.

#### 2.3 Flux Calculations

# 2.3.1 Fco2

CO<sub>2</sub> flux was calculated from the 10 Hz data as  $F_{CO2} = p_{\overline{a}} \text{ w}^* c^*$ , where  $p_{\overline{a}}$  (mol m<sup>-3</sup>) is the mean dry air density, w (m s<sup>-1</sup>) is the vertical wind speed, c is the CO<sub>2</sub> mixing ratio (µmol mol<sup>-1</sup>), primes indicate fluctuations about the mean, and the overbar corresponds to the time average (20 minutes for this study). As the product of measurements from different instruments, the accuracy of the  $F_{CO2}$  measurement is challenging to quantify without an independent validation, which was not performed. The L1-7200 has a measurement accuracy of  $\pm 1\%$  with an RMS noise of 0.11 ppm at 10 Hz, while the vertical wind speed of the CSAT3 is accurate within  $\pm 0.04$  m s<sup>-1</sup> with an RMS noise of 0.0005 m s<sup>-1</sup>. While the noise can occasionally be larger than the true environmental fluctuations, it has been found to minimally influence the calculated  $F_{CO2}$  because the noise from the separate instruments is uncorrelated and therefore filtered out by the flux calculation (Miller et al., 2010).

An investigation of  $F_{CO2}$  measurement uncertainties from ships indicated a detection limit for a dried, closed-path eddy covariance system of roughly  $|\Delta pCO_2| > 35$  µatm for the mean wind speed observed in this study (Blomquist et al., 2014). The  $\Delta pCO_2$  in the region often exceeds this value (Duke et al., 2021; Sims et al., 2023). Additionally, we expect some reduction in the detection limit (i.e., increased sensitivity) for this study compared to ship-based studies, because the measurements were from a stationary tower. Therefore, the observations avoid some common sources of uncertainty experienced from moving

Deleted: et al.

platforms, such as the needed for a complex wind vector motion correction and tilt effects that degrade the performance of the LI-7200 (Miller et al., 2010; Vandemark et al., 2023).

While we cannot perform a direct assessment of  $F_{CO2}$  uncertainty, we can estimate the order of magnitude of the uncertainty by assessing the variation in  $F_{CO2}$  measurements during periods expected to have stable fluxes. Here we do that by calculating the standard deviation for 6-hour intervals during periods of full ice cover, when diurnal variations in  $F_{CO2}$  were expected to be minimal. The standard deviation across these winter periods had a mean of  $\pm 1.02$  mmol m<sup>-2</sup> d<sup>-1</sup> and a median of  $\pm 0.75$  mmol m<sup>-2</sup> d<sup>-1</sup>. Spring and summer seasons were excluded from the estimate because standard deviation measured during those periods was expected to be a combination of measurement uncertainty and actual diurnal  $F_{CO2}$  trends.

# 195 2.3.2 pCO<sub>2</sub>

The difference in partial pressure of  $CO_2$  ( $pCO_2$ ) between the air and the water dictates the direction of the flux (up or down), while the gas transfer velocity (k) describes the efficiency of transport. The latter incorporates all of the physical processes at the air-sea interface that affect gas exchange. Using existing parameterizations for k we can use the  $F_{CO2}$  measurements to estimate the partial pressure of  $CO_2$  in water ( $pCO_{2w}$ ). Viewing the flux data as  $pCO_2$  provides context for the flux by showing the seasonal pattern of waterside carbon inventories, without the short-term variability caused by the impact of wind speed on k and therefore  $F_{CO2}$  magnitude. To estimate  $pCO_{2w}$  we set our measured  $F_{CO2}$  equal to the open water bulk formula for  $CO_2$  flux:

$$F_{CO2} = k s \left[ pCO_{2w} - pCO_{2air} \right], \tag{1}$$

where k is estimated from wind speed using Wanninkhof (2014), s is the solubility of CO<sub>2</sub> in seawater (calculated using satellite-derived sea surface temperature [SST] and salinity [S<sub>SW</sub>] data from the mooring), and pCO<sub>2air</sub> was partial pressure of CO<sub>2</sub> in air. Because pCO<sub>2w</sub> was the single unknown in the equation we were able to solve for it.

The processes affecting Fco2 from sea ice are different from open water, but a similar bulk flux formula can be applied. During periods of full sea ice cover, we can use this formula to estimate the partial pressure of CO<sub>2</sub> in ice (pCO<sub>2ice</sub>). This value describes the concentration of CO<sub>2</sub> in the ice, which in this context could represent any ice surface interacting with the atmosphere including snow crystals, the sea ice surface, or the sea ice volume (including brine). Like pCO<sub>2w</sub> in open water, pCO<sub>2ice</sub> in ice can vary over time and its difference from pCO<sub>2nir</sub> is still expected to dictate the direction of flux. In the laboratory study of Kotovitch et al. (2016), Fco<sub>2</sub> was measured in a tank over periods of forming, thickening, and melting sea ice.

Supporting measurements of pCO<sub>2air</sub> and pCO<sub>2ice</sub> enabled the derivation of a gas transfer coefficient (K<sub>ice</sub>) using the following bulk formula:

$$F_{CO2} = K_{ice} \left[ pCO_{2ice} - pCO_{2air} \right]_{\Psi}$$
 (2)

- The  $K_{lce}$  parameter encapsulated both the gas transfer velocity and solubility of CO<sub>2</sub> in ice. This was done to avoid estimating solubility using seawater-based functions of temperature and salinity outside the range for values for which they were designed.  $\underline{K}_{lce}$  during periods of ice growth was 2.5 mol m<sup>-2</sup> d<sup>-1</sup> atm<sup>-1</sup>, while for periods of ice decay it was 0.4 mol m<sup>-2</sup> d<sup>-1</sup> atm<sup>-1</sup> (Kotovitch et al., 2016).
- Because we did not collect in situ pCO<sub>2ice</sub> measurements we could not use Eq. (2) to calculate K<sub>ice</sub> for independent verification.
  Instead, we estimated pCO<sub>2ice</sub> during periods of full ice cover using Eq. (2) with measured F<sub>CO2</sub> and pCO<sub>2air</sub> and the K<sub>ice</sub> values for ice growth and decay found by Kotovitch et al. (2016). Comparisons of estimated pCO<sub>2ice</sub> to previous in situ measurements were used to determine if the laboratory-derived K<sub>ice</sub> values were applicable in field conditions.
- For periods where the surface was a mix of open water and sea ice we estimated pCO<sub>2w</sub> by scaling F<sub>CO2</sub> linearly to the fraction of open water (f = 1 SIC) in front of the tower (Butterworth and Miller 2016). In these cases, we omitted the influence of air-ice gas exchange in the calculation of pCO<sub>2w</sub> due to the fact that K<sub>ice</sub> is much lower than its equivalent (k s) for the air-water interface (Wanninkhof 2014; Kotovich et al., 2016). Under the environmental conditions (e.g., temperatures, salinity, etc.) in this study we estimated that F<sub>CO2</sub> from open water is roughly 20 times more efficient. Therefore, the omission of air-ice gas exchange is expected to have a minimal influence on the pCO<sub>2w</sub> calculation. For this work, seasons were defined by

spring being broadly represented by ice melt, summer by open water, and fall by ice formation.

in situ observations (i.e., by visits to the station, and from camera images) rather than standard astronomical definitions, with

3 Results & Discussion

# 240 3.1 Meteorology

The meteorological conditions during the two measurement years followed similar trends. Air temperatures rose above the freezing point of seawater in late May/early June and remained positive until September when they dropped below freezing again (Fig. 3). As expected, the timing of sea ice melt in the spring and freeze-up in the fall coincided with the timing of these temperature transitions. The period of the spring melt (from initial melt to fully open water) lasted roughly seven weeks in both years. In contrast, the freeze-up period (from first freeze to full ice cover) in 2017 lasted four weeks, though additional thickening was presumed to be occurring following the formation of landfast ice. Wind speeds were low to moderate at 6.1 ±

**Deleted:** In the laboratory tank sea ice study of Kotovitch et al. (2016) measurements of  $F_{CO2,p}CO_{2uir}$ , and  $pCO_{2iev}$  were used to determine  $K_{lcc}$ —a parameter that encapsulates both the gas transfer velocity and solubility of  $CO_2$  in ice. Here, we estimate  $pCO_{2iev}$  during periods of full ice cover by setting our measured  $F_{CO2}$  equal to the equation

Deleted: .

**Deleted:** where K<sub>ice</sub> was the gas transfer velocity for ice growth and decay (2.5 and 0.4 mol m<sup>-2</sup> d<sup>-1</sup> atm<sup>-1</sup> respectively) found by Kotovitch et al. (2016).

Deleted:

Figure 3. Mean meteorological conditions relevant to Fco2 including (a,b) sea ice concentration (SIC), (c,d) air temperature, (e,f) wind direction, and (g,h) wind speed. All data are 6-hour averages except AMSR-2 SIC which is a daily mean (Spreen et al., 2008) and the spring 2017 portion of the Camera SIC data which is intermittent. Seasonal date ranges from Table 1 are illustrated by the color band on the top of the figure with sub-seasons early, mid, late labeled as E, M, L. The red band on (e,f) indicates southerly wind sector (150° – 210°) discarded for flux analysis.

Table 1. Date ranges of seasonal transitions in 2017 and 2018. Fco<sub>2</sub> direction refers to whether the season was characterized by outgassing (+) or uptake (-); while " $\Delta$  in Fco<sub>2</sub>" refers to whether fluxes were increasing (†) or decreasing ( $\downarrow$ ) across the season. Note that for the late summer period "fully mixed" indicates that the water was mixed down to the nearby 39-m deep mooring. Seasonal cutoff dates were determined by transition to different defining processes, as identified by in situ observations from site visits and camera images.

|        | Period | Defining Process        | Dates (2017)  | Dates (2018)  | F <sub>CO2</sub> direction | $\Delta$ in $F_{CO2}$ |
|--------|--------|-------------------------|---------------|---------------|----------------------------|-----------------------|
|        |        |                         |               |               | (0, -, +)                  | (0,↓,↑)               |
| Spring | Early  | Freeze-thaw             | 05/21 - 06/11 | 06/09 - 06/15 | 0                          | 0                     |
|        | Mid    | Melt Ponds              | 06/11 - 06/25 | 06/15 - 07/05 | -                          | <b>1</b>              |
|        | Late   | Break-up                | 06/25 - 07/13 | 07/05 - 07/26 | -                          | <b>1</b>              |
|        | Early  | Pre-peak SST            | 07/13 - 08/13 | 07/26 -08/13  | -,+                        | 1                     |
| Summer | Mid    | Post-peak SST           | 08/13 - 10/01 | 08/13 - NA    | +,-                        | 1                     |
|        | Late   | Fully mixed             | 10/01 - 10/14 | NA            | _                          | 1                     |
| E 11   | Early  | Ice formation           | 10/14 - 11/14 | NA            | -,+                        | 1                     |
| Fall   | Late   | Thickening landfast ice | 11/14 - 12/31 | NA            | +                          | 1                     |
| Winter |        | Solid landfast ice      | 12/31 -       | NA            | 0                          | 0                     |

#### 275 3.2 Annual Fluxes

The direction of F<sub>CO2</sub> (sink vs. source) varied seasonally (Fig. 4). During spring, mid-to-late summer, and early fall the region acted as a sink, while during the early summer and late fall it acted as a source. Over the course of 2017 the fluxes from the separate seasons nearly balanced out, with the total annual flux being only 6% of the absolute flux (Table 2).

| Deleted: e      |  |
|-----------------|--|
| Deleted: f      |  |
| Deleted: et al. |  |

| Deleted: Flux |  |
|---------------|--|
| Deleted: flux |  |

| -( | Deleted: Flux   |
|----|-----------------|
| (  | Deleted: flux   |
| Y  | Formatted Table |

Figure 4. Six-hour average  $F_{CO2}$  (mmol m<sup>-2</sup> d<sup>-1</sup>) for years (a) 2017 and (b) 2018. Color represents sea ice concentration. Black curve represents a locally-weighted least-squares regression line fit with a quadratic polynomial. <u>Uncertainty in the  $F_{CO2}$  measurement was quantified by calculating the standard deviation from each 6-hour average (comprised of eighteen 20-minute flux intervals) during periods of full ice cover, when diurnal  $F_{CO2}$  variations were minimal. The standard deviations across these winter periods had a mean of  $\pm 1.02$  mmol m<sup>-2</sup> d<sup>-1</sup> and a median of  $\pm 0.75$  mmol m<sup>-2</sup> d<sup>-1</sup>.</u>

Table 2. Seasonal measurements of Fco2, presented as cumulative fluxes and percentage of annual flux for 2017. The cumulative fluxes were calculated by integrating the area under the local regression curve from Figure 4 between the zero crossings separating periods of uptake from periods of outgassing. In this instance only, the use of terms "Spring", "Summer", and "Fall" are defined based on these zero crossings, identified in the "Dates" column of the table. Note that they are not precisely aligned with seasonal demarcations defined in Table 1 (which are used in all subsequent analyses). This was done to avoid integrating using seasonal demarcations that straddled positive and negative flux transitions.

|                   | Dates         | Total F <sub>CO2</sub> | Seasonal |
|-------------------|---------------|------------------------|----------|
|                   | (2017)        | (g-C m <sup>-2</sup> ) | Annual   |
| Spring uptake     | 05/25 - 07/22 | -0.7                   | 13.5%    |
| Summer outgassing | 07/22 - 09/08 | 1.7                    | 33%      |
| Summer uptake     | 09/08 - 10/28 | -2.1                   | 40%      |
| Fall outgassing   | 10/28 - 12/28 | 0.7                    | 13.5%    |
| Total             |               | -0.3                   | 6%       |
| Total             |               | 5.2                    | 100%     |

# 3.3 Spring

#### 3.3.1 Spring Results

For this study, we mark the beginning of the spring season as the moment when mean daytime temperature rises above 0°C (Fig. 3c,d). In the two years presented, this spring start date shifted by about 3 weeks. This difference appeared to play a role

**Deleted:** Note that in this instance only the use of terms spring, summer, and fall are defined based on the zero crossings of the local regression curve from Figure 3 (and therefore straddle the seasonal demarcations defined in Table 1). The cumulative fluxes in this table were calculated by integrating the area under this curve.

Deleted: lux
Formatted: Subscript

in the differences in CO<sub>2</sub> flux direction and magnitude throughout the remainder of each season, which will be discussed in Section 3.4.1.

- The spring season is marked by distinct periods (Table 1). In early spring, the surface is characterized by freeze-thaw cycles (e.g. Hanesiak et al., 1996). While there may be leads during this period, the ice is landfast, with typically 100% coverage. During mid spring, standing water melt ponds form on the ice surface, still with 100% ice coverage. The late spring season is marked by a break-up of the sea ice, where the ice concentration decreases from 100% to 0% coverage.
- During early spring the behavior of snow melt / refreezing appears to be the key factor affecting F<sub>CO2</sub>. On early spring days in which air temperature oscillated around the melting point, F<sub>CO2</sub> oscillated with a mean range of 1 mmol m<sup>-2</sup> d<sup>-1</sup> on a diurnal cycle negatively correlated with air temperature (Fig. 5b). During the day, positive temperatures caused melt, resulting in a negative F<sub>CO2</sub> (uptake). At night, when negative temperatures caused water to refreeze and expel CO<sub>2</sub> gas, F<sub>CO2</sub> was positive (outgassing). Incoming solar radiation did not have an immediate impact on F<sub>CO2</sub> (Fig. 5a,b), though was correlated once lagged to temperature. During this period the sea surface was characterized by an average ice coverage of 99%. In photographs from the 19 days included in the freeze—thaw analysis, the surface showed a slight darkening during the daytime, consistent with Hanesiak et al. (1996) who observed diurnal albedo patterns caused by increased water content during the day, and freezing overnight. At this time, no discernable standing water melt ponds had formed.
- We estimated pCO<sub>2ice</sub> using Eq. (2) and found a diurnal range of 600 μatm during this period, corresponding in sign to the direction of the flux (Fig. 5b,c). Mean diurnal minimum pCO<sub>2ice</sub> was roughly 0 μatm and occurred in the afternoon, coinciding with the warmest air temperatures and greatest active melting. The mean diurnal maximum was 600 μatm and occurred shortly after sunrise, when mean air temperature was at a minimum at -2°C.

Deleted: 1

Figure 5. Average diurnal cycle of (a) incoming shortwave radiation, (b) mean  $F_{CO2}$  and air temperature, and (c)  $pCO_{2ice}$  and  $pCO_{2air}$  for the 19 spring days which oscillated between positive and negative air temperatures. Shaded areas represent 1.96 × standard error (i.e., the 95% confidence interval).

As temperatures increased during mid spring (Table 1), standing water melt ponds began to form on the landfast ice. During this period the magnitude of Fco2 increased, showing more strongly negative fluxes (i.e., uptake), with occasional outgassing events (Fig. 4a,b). The positive to negative Fco2 diurnal oscillations seen in early spring (Fig. 5b) were no longer evident (Fig. 6).

Figure 6. Mean diurnal  $F_{\rm CO2}$  and air temperature during mid spring (11 June 2017 – 25 June 2017; 15 June 2018 – 5 July 2018). The blue and red shaded regions represent 1.96 × standard error (i.e., the 95% confidence interval) of  $F_{\rm CO2}$  and T, respectively. The dotted green line represents median  $F_{\rm CO2}$  and the green shaded region represents the 25th to 75th percentiles of  $F_{\rm CO2}$ .

During late spring (Table 1) the landfast ice begins to break up and the ocean surface in front of the tower is characterized by varying concentrations of sea ice in the form of ice floes. During this period (lasting several weeks)  $F_{CO2}$  becomes even more strongly negative. This increased  $CO_2$  uptake was likely due to the exposure of seawater that had low  $pCO_{2w}$  relative to atmospheric  $pCO_{2wi}$ . The  $pCO_{2w}$  (calculated using Eq 1) decreased from a mean of 394  $\mu$ atm ( $\Delta pCO_2$  of -10  $\mu$ atm) during mid spring to 373  $\mu$ atm ( $\Delta pCO_2$  of -29  $\mu$ atm) during late spring (Fig. 7). This decrease in  $pCO_{2w}$  acts in opposition to the water temperature effect on  $pCO_{2w}$  during this period. In both years, water temperature (both satellite SST and 13-m mooring) increased by roughly 1°C over the period, which independently should cause a roughly 20- $\mu$ atm increase in  $pCO_{2w}$ , based on the direct positive relationship between water temperature and  $pCO_{2w}$  (Takahashi et al., 1993).

Deleted: 2002

Figure 7. Time series of  $\Delta p CO_2$  (i.e.,  $p CO_{2w} - p CO_{2air}$ ) estimated using Eqs. 1 and 2. Color of the line represents ice concentration. The black curve represents a locally-weighted least-squares regression line fit with a quadratic polynomial.

#### 3.3.2 Spring Discussion

Springtime  $F_{CO2}$  is characterized by the distinct physical processes related to freeze—thaw, melt ponds, and ice break-up. These processes likely all occur to some degree throughout the spring period, but they generally progress sequentially along with the advance of warming over the spring.

The observation of diurnal cycles in early spring influenced by active melting and freezing has implications for sampling design for instruments not intended for continuous deployment (e.g., chambers) – namely that measurement biases could arise based on collection time (e.g., cold morning measurements would predict a  $CO_2$  source and warm afternoon measurements would predict a sink). Despite the diurnal variability, the ice acts as a weak sink during this period with mean flux of -0.35 mmol  $m^{-2}$  d<sup>-1</sup>.

While the negative mean  $F_{CO2}$  suggested mean  $pCO_{2ice}$  was below mean  $pCO_{2iir}$ , the diurnal oscillations in  $F_{CO2}$  indicated diurnal changes in the magnitude of  $pCO_{2ice}$ . The diurnal range of  $pCO_{2ice}$  was quite large (0 – 600  $\mu$  matrix; Fig. 5c) and was likely due to physical processes associated with the phase change of water. That is, the expulsion of  $CO_{2}$  gas as water freezes and then the subsequent melting of low  $pCO_{2ice}$  (Nomura et al., 2006; Rysgaard et al., 2011; Kotovitch et al., 2016). During active melting we found a diurnal minimum in our flux-estimated  $pCO_{2ice}$  of 0  $\mu$  matrix, which corresponds in magnitude to previous directly-measured, in situ melt pond  $pCO_{2}$  of 36  $\mu$  matrix (Geilfus et al., 2015). The low  $pCO_{2}$  of melt ponds are expected to immediately begin to equilibrate toward atmospheric values (Geilfus et al., 2015). However, the diurnal change in flux direction from uptake to outgassing indicates that  $pCO_{2ice}$  rose above atmospheric values. This suggests that the  $CO_{2}$  gas

Deleted: et al.

Deleted: et al.

Deleted: et al.

Deleted: et al.

expelled during freezing accumulated in a thin, supersaturated layer near the surface. This is in line with the laboratory experiment of Kotovich et al. (2016), who also observed outgassing during freezing due to supersaturation in the top 5 cm of ice, while the underlying water remained undersaturated with respect to the atmosphere.

The large range of  $pCO_{2ice}$  in this study has some analogies in the literature. This includes the range measured by Delille et al. (2014) in Antarctic pack ice (roughly  $50 - 900 \mu atm$ ) and the range observed by Geilfus et al. (2015) in Arctic springtime ice ( $36 - 380 \mu atm$ ). While these studies represent daytime-only  $pCO_{2ice}$  measurements over longer time frames (seasonal and sub-week, respectively), they show that  $pCO_{2ice}$  of these magnitudes ( $0 - 600 \mu atm$ ) are plausible. The agreement between our estimated  $pCO_{2ice}$  and previous direct in situ measurements of  $pCO_{2ice}$  suggests that the gas transfer coefficient for melting ice measured by the laboratory experiment of Kotovitch et al. (2016; which we used to estimate  $pCO_{2ice}$  from our flux measurements) may be reasonably applicable to the real-world environment. However, it is worth noting that  $pCO_{2ice}$  (Fig. 5c) occasionally dropped below zero, which is a physically impossible value. Such instances may indicate that the  $K_{ice}$  value used to calculate  $pCO_{2ice}$  was too small. Because  $K_{ice}$  combines both gas transfer velocity and solubility, inaccuracies in either term could be responsible. However, it is also possible that the negative values of  $pCO_{2ice}$  are simply due to the random error inherent in eddy covariance systems. Because random error can cause both positive and negative deviations in measured flux, these data points were retained to avoid biasing the average.

measurements. We assumed that  $pCO_{2ice}$  was zero during periods of time in early spring when temperatures were positive. This represents the lowest possible  $pCO_{2ice}$  and therefore the most negative  $\Delta pCO_2$  that was physically possible at the site. Using Eq. (2) with this prescribed  $pCO_{2ice}$  and mean  $F_{CO2}$  we calculated a  $K_{melt}$  value of 0.36 mol m<sup>-2</sup> d<sup>-1</sup> atm<sup>-1</sup>. This is nearly identical to the  $K_{melt}$  of 0.4 mol m<sup>-2</sup> d<sup>-1</sup> atm<sup>-1</sup> found by Kotovitch et al. (2016). This is a rough estimate for several reasons. First, the  $pCO_{2ice}$  is not expected to be zero for this entire period. Past studies (e.g., Geilfus et al., 2015) have shown that  $pCO_2$  of freshly melted ice approaches zero, but that value is expected to rise quickly as the water equilibrates with the atmosphere. A  $pCO_{2ice}$  value of zero is therefore theoretically possible for a rapidly melting surface, but it would be a transient state. An average  $pCO_{2ice}$  higher than zero would result in a higher  $K_{melt}$ . Secondly, this calculation assumes that 100% of the surface is decaying ice – which may not be true. With a lower fraction of the surface actively decaying we expect the estimated  $K_{melt}$  to increase. Overall, however, it provides a constraint on the lower limit of  $K_{melt}$  and suggests that the laboratory value proposed by Kotovitch et al. (2016) is the correct order of magnitude in the natural environment. That the laboratory value aligns with the lower limit measured in the field makes sense, given that some of the natural factors that are known to increase fluxes (e.g., wind) are absent in laboratory settings.

To further constrain the gas transfer coefficient over melting sea ice (Kmelt) we ran an additional test using our flux

Mid spring (Table 1) was characterized by the formation of large standing melt ponds on the landfast ice (Fig. 2b). During this period, we observed a discontinuation of the diurnal cycles observed during early spring (i.e., negative correlation between

Deleted: Additionally, t

Deleted: estimated

Deleted: matches that

Deleted: at

Deleted: y

Deleted: s

Deleted: s

Deleted: s

Deleted: seasonal changes in pCO<sub>2ics</sub>, it shows

Deleted: 1

Deleted:

Fco2 and temperature). This was likely due to consistently positive air temperatures eliminating the potential for refreezing, ending freeze—thaw related forcings on the flux. Mid spring also had more strongly negative  $F_{CO2}$  than early spring. This suggests that melt ponds were acting as a sink for  $CO_2$ . This is in line with previous studies which have found that low  $pCO_{2w}$  concentrations in melt water cause melt ponds to be a net sink of  $CO_2$  (Semiletov et al., 2004, Geilfus et al., 2015).

concentrations in melt water cause melt ponds to be a net sink of CO<sub>2</sub> (Semiletov et al., 2004, Geilfus et al., 2015).

A quantitative analysis of pCO<sub>2</sub> during the melt pond period was not attempted due to uncertainties in gas transfer coefficients. The laboratory-derived K<sub>ice</sub> values for ice growth and decay that were applied during the freeze—thaw period were not expected

ponds are exchanging gas with the atmosphere with physics more closely aligned to air-water gas exchange than air-ice gas exchange, there are reasons to believe that open ocean parameterizations of gas transfer velocity are not entirely suitable to melt ponds, due to the expected differences in wind-wave fields and waterside turbulence between the two environments.

to be applicable over flux footprints that contained both ice and standing water melt ponds. And while we assume that melt

During the transition to ice breakup in late spring we measured consistently negative F<sub>CO2</sub>, which indicated pCO<sub>2w</sub> values during this period were below atmospheric values. Because increasing water temperatures during this period should have led to increased pCO<sub>2w</sub>, the observed decrease in pCO<sub>2w</sub> suggests that other processes were driving the low pCO<sub>2w</sub> values observed during this period. For example, hyperspectral transmitted irradiance measurements made in spring 2017 on the nearby mooring revealed ice algal and under-ice phytoplankton blooms occurring from 5 March to 21 May and 1 to 10 June, respectively (Yendamuri et al., 2024), that could have drawn down pCO<sub>2w</sub>. However, primary production in the area is relatively low compared to other Arctic regions due to nitrogen limitation (Kim et al., 2021; Back et al., 2021) and thus, may not have significantly contributed to the low pCO<sub>2w</sub> observed. An alternative process is simply ice melt, which has been shown to lower pCO<sub>2</sub> both through simple mixing of low-pCO<sub>2</sub> melt water, and due to non-linearities in carbonate system chemistry (Yoshimura et al, 2025). The salinity mooring data was inspected to determine whether melt water dilution was observed. At

13 m depth there was a small decrease in  $S_{SW}$  (-0.2) over the late spring period. This would correspond to a small decrease in  $pCO_{2w}$  (-3  $\mu$ atm), a relatively small  $F_{CO2}$  forcing. However, because the water was stratified at this period (i.e.,  $SST > T_{13m}$ ), it is possible (and likely) that the change in  $S_{SW}$  at the surface was greater, resulting in a larger  $F_{CO2}$  forcing.

# 3.4 Summer

#### 460 3.4.1 Summer Results

Here we define summer as the open water period, which spans from mid-July to mid-October (Table 1). In 2017, the difference between SST (derived from satellite) and 13-m water temperature (T<sub>13m</sub>, from the mooring) showed that the sea was stratified from May to August (Fig. 8). On Aug. 13 the SST peaked for the season at 9.8°C. For the remainder of August, SST decreased while T<sub>13m</sub> increased, indicating a growing mixed layer, which reached 13-m depth on Sep. 2 when equivalence between SST and T<sub>13m</sub> was reached. The two temperatures tracked together until early October when sea ice began to form. Temperature measurements obtained at 22 and 39 m depths showed that by Oct. 1 the water in the region became mixed from at least the

Deleted: et al.

Deleted: et al.

Deleted: et al.

Deleted: et al.

surface to a 39-m depth, which is close to the charted bottom depths for most of the area within the flux footprint. A similar story unfolded in 2018, with SST also reaching its peak on Aug. 13. However, compared to 2017 its maximum temperature was much lower at 4.4°C, presumably due to the delayed onset of melt, providing a shorter window for the absorption of incoming solar radiation by the sea surface. Because the mooring data stopped on Aug. 14 mixed layer depths during the second half of summer 2018 were not available.

Figure 8. Shows time series of smoothed  $F_{CO2}$  (local regression line from Fig. 4) with color indicating sea ice concentration, SST from AVHRR (light blue line), and water temperature at 13-m depth from the mooring (gray line).

Figure 8 shows the  $F_{CO2}$  dependence on water temperature as it varies across seasons. During the open water period in 2017 the mean  $F_{CO2}$  tracks with SST, rising together in July and August, peaking in mid-August, and decreasing through late August to October. We separated the summer season into three subseasons (early, mid, and late) corresponding to changes in environmental conditions. The early season (13 July 2017 – 13 Aug 2017) was from the beginning of open water until peak SST and showed increasing  $F_{CO2}$  (Fig. 8). The mid season (13 Aug 2017 – 1 Oct 2017) was from peak SST until the water profile became unstable and showed decreasing  $F_{CO2}$  (Fig. 8). The late season (1 Oct 2017 – 14 Oct 2017) was the period immediately preceding the onset of freezing in which the mixed layer deepened. We then investigated the role of thermodynamic processes on the observed seasonal  $F_{CO2}$  changes. Figure 9 shows a  $pCO_{2w}$  estimate derived from  $F_{CO2}$  using  $F_{CO2}$  using  $F_{CO2}$  projection calculated using established temperature and salinity relationships  $\frac{1}{pCO_{2w}}$   $\frac{apcO_{2w}}{assT} \approx 0.0423^{\circ}C^{-1}$ 

from Takahashi et al. [1993];  $\frac{S_{SW}}{pCO_{2w}} \frac{\partial pCO_{2w}}{\partial S_{Sw}} \approx 1$  from Sarmiento and Gruber [2006]). For the thermodynamic projection, the F<sub>CO2</sub>-derived  $pCO_{2w}$  estimate for the first day of early summer was used as a starting  $pCO_{2w}$ , then projected forward for each flux interval through the end of summer using only the above SST and S<sub>SW</sub> relationships. In both early and mid summer, the two  $pCO_{2w}$  estimates track well, indicating that changes in SST and S<sub>SW</sub> are important drivers of F<sub>CO2</sub> changes during these seasons. In late summer, the curves show greater divergence with the F<sub>CO2</sub>-derived  $pCO_{2w}$  estimate showing larger values (over

Deleted: seasonal temperature dependence of

25 μatm greater) than the thermodynamic projection. While the F<sub>CO2</sub> increased from its seasonal low during this late summer period (Fig. 4 & 8; due to the reduced wind speed [Fig. 3e]), the F<sub>CO2</sub>-derived *p*CO<sub>2w</sub> estimate continued to drop in magnitude (Figs. 7 & 9). This was in opposition to the SST forcing, but coincided with deepening of the mixed layer and increased S<sub>SW</sub> values.

Figure 9. Summer 3-day average time series of  $pCO_{2w}$  derived from  $F_{CO2}$  using Eq. (1) (black line) and  $pCO_{2w}$  projection calculated using temperature and salinity relationships (Takahashi et al., 1993; Sarmiento and Gruber, 2006; blue line). Shaded regions represent standard deviation.

The overall pattern of Fco2 in 2018 was similar to 2017, with downward fluxes predominating during spring melt and breakup, then increasingly upward fluxes as SST increased during early summer (26 July 2018 – 13 Aug 2018). Like 2017, Fco2 510 began to decrease as soon as the maximum SST was reached at the start of mid summer (13 Aug 2018 – N/A). However, the first two weeks of September showed a turn towards increasingly positive fluxes (Fig. 4b, Fig. 7b). In contrast, during this same period in 2017 the fluxes were becoming increasingly negative.

#### 515 3.4.1 Summer Discussion

In summer, thermodynamic drivers appear to be the most important contributors to the direction and magnitude of  $F_{CO2}$ . For most of the summer, the trend in  $F_{CO2}$  corresponds to the trend in SST. Both increase in early summer, both decrease in mid summer (Figs. 4 & 8). The mechanism causing this pattern is the direct positive relationship between SST and  $p_{CO2}$  where  $p_{CO2}$  is the summer of  $p_{CO2}$  in the s

**Deleted:** Separating the summer season into an early season (13 July 2017 – 13 Aug 2017; corresponding to the beginning of open water until peak SST) and a mid season (13 Aug 2017 – 1 Oct 2017; peak SST until mixed layer deepens), we find the early summer season shows a shallower (weaker) relationship between Fco2 and SST than the mid season (Fig. 9). Interestingly, in late summer (1 Oct 2017 – 14 Oct 2017; the period immediately preceding the onset of freezing) Fco2 increases in opposition to the SST forcing, but coincides with deepening of the mixed layer.

 $\label{eq:Deleted: Fco2} \begin{tabular}{l} \textbf{Deleted:} Fco2 (mmol m² d⁻¹) plotted against SST (°C) as an early season (13 July 2017 – 13 Aug 2017) component (purple) and a late season (13 Aug 2017 – 1 Oct 2017) component. The corresponding straight lines through the data points represent the least-squares fit.$ 

(Takahashi et al., 1993), As SST increases, it causes pCO<sub>2w</sub> to increase, which results in increased outgassing of CO<sub>2</sub> to the atmosphere. Ssw also has a direct positive relationship with pCO<sub>2w</sub> (Sarmiento and Gruber, 2006). In this instance, steady reductions in S<sub>SW</sub> over the course of the early and mid summer periods (28 down to 25) partially offsets the projected peak 535 magnitude of pCO<sub>2w</sub> by the SST effect alone. The projection of pCO<sub>2w</sub> using both SST and S<sub>SW</sub> effects tracks well with the F<sub>CO2</sub>-derived pCO<sub>2w</sub> estimate (Fig. 9). This suggests that SST and S<sub>SW</sub> are the main drivers of changes to F<sub>CO2</sub> in the early and mid summer periods. The one period of the summer in which the thermodynamic pCO<sub>2w</sub> projection most noticeably diverges from the F<sub>CO2</sub>-derived pCO<sub>2w</sub> estimate is late summer (1 Oct 2017 – 14 Oct 2017). During this period the SST continues to drop, but S<sub>SW</sub> begins to increase (25 up to 27). This corresponds to a reduced (but still negative) slope to both the F<sub>CO2</sub>-derived pCO<sub>2w</sub> estimate and the thermodynamic projection. The cause of the increased S<sub>SW</sub> was the reversal of the temperature profile from stable to unstable (i.e.,  $SST < T_{13m} < T_{22m} < T_{39m}$ ) resulting in greater upward mixing of higher salinity water from depth. While the similar trends in both the pCO<sub>2w</sub> estimate and the thermodynamic projection suggest that SST and S<sub>SW</sub> are still important drivers of F<sub>CO2</sub> during late summer, the higher magnitudes of the pCO<sub>2w</sub> estimate compared to the thermodynamic projection suggest an additional source of increased  $pCO_{2w}$ . One possibility is that the increased mixing of water from depth during this late summer period may have, in addition to increasing Ssw, brought CO2-rich waters to the surface, thus slightly offsetting some of the  $pCO_{2w}$  reductions expected by the thermodynamic processes alone.

As stated above, the pattern of F<sub>CO2</sub> in 2018 was similar to 2017, with the exception of 2018 showing increasing positive fluxes and increasing pCO<sub>2w</sub> in the first two weeks of September, running in opposition to the SST forcing. One explanation is that the lower SST during 2018 enabled mixed layer deepening a month earlier than the previous year, causing mixing to increase pCO<sub>2w</sub> (e.g., due to S<sub>SW</sub> and CO<sub>2</sub> concentration effects) earlier in the season. Unfortunately, the mooring temperature and salinity data were not available during this period to confirm. However, an inspection of the flux cospectra during this period showed no reason to discount this upward trend on the grounds of flux measurement error.

3.5 Fall

# 3.5.1 Fall Results

The fall season was defined by the occurrence of sea ice formation. Photographs from the camera at the top of the tower confirmed that freezing began on 14 Oct 2017 and continued to increase until consistent, full ice cover was reached on 14 Nov 2017. This initial freeze-up period was defined as early fall, followed by a late fall period of thickening landfast ice, during which the site continued to measure active Fco2. At the outset of freezing, the fluxes were downward (uptake), but transitioned upward (outgassing) shortly after ice formation (28 Oct 2017). They remained upward until they reached zero at the end of December 2017.

Deleted: In summer, as during spring, both water temperature and biological activity appear to be important contributors to the direction and magnitude of Fco2. For most of the summer the trend in F<sub>CO2</sub> corresponds to the trend in water temperature. Both increase in early summer, both decrease in mid summer (Fig. 4). The mechanism causing this pattern is the direct positive relationship between water temperature and pCO<sub>2w</sub> (Takahashi et al., 2002). As temperature increases, it causes pCO<sub>2w</sub> to increase, which results in increased outgassing of CO2 to the atmosphere. However, the temperature effect does not appear to be the only mechanism influencing the fluxes, which is made clear by the different Fco2 vs SST relationships in early and mid summer (Fig. 9). This difference is most likely due to the tendency for this region to experience most of its primary production (both ice algal blooms, and under-ice phytoplankton blooms) early in the season. The photosynthesizing organisms draw down aqueous pCO2w, reducing the degree of outgassing that would be expected with no photosynthesis occurring After SST peaks in early August the relationship between Fc02 and SST (positive relationship with both decreasing) is greater, since the biological activity during this period is reduced due to lower insolation, diminished nutrients, and deeper mixing. The one period of the summer in which the positive relationship between Fco2 and water temperature breaks down is late summer (1 Oct 2017 - 14 Oct 2017). Here Fco2 increases while water temperature continues to drop. During this period there is not expected to be much biological activity to force Fco2 up (e.g., respiration) or down (e.g., photosynthesis). However, it is coincident with the timing of the mixed layer deepening, which may have brought CO2-rich waters to the surface. This would be expected to increase Fco2 in this late summer period. However, the main cause of this trend reversal appears to be a reduction in wind speed during this period (Fig. 3e), rather than biological or oceanographic forcings. This physical (wind) explanation for the reduced Fco2 is confirmed by the pCO2w estimate (which is calculated with measured Fc02 and wind speed using Eq. (1)) continuing to drop during this period (Fig. 3e). This suggests that pCO2w is still primarily controlled by the water temperature relationship.

As stated above, the pattern of Fco in 2018 was similar to 2017. with the exception of 2018 showing increasing positive fluxes and increasing pCO<sub>2w</sub> in the first two weeks of September, running in opposition to the water temperature forcing. One explanation is that because the temperature trends were notably less steep in 2018 compared to 2017, other forcings may have played more prominent roles. An inspection of the flux cospectra during this period showed no reason to discount this upward trend on the grounds of measurement error. While the mooring temperature data was not available during this period to confirm, we expect that the lower SST during 2018 enabled mixed layer deepening a month earlier than the previous year, causing mixing to increase pCO2w earlier in the season.

The downward Fco2 at the beginning of the fall occurred when sea ice concentrations were lowest. This likely represents dominant flux between the atmosphere and open water areas, since  $\Delta p CO_2$  between the water and air was negative at the onset of freezing. The upward fluxes that follow this period (Nov – Dec) coincide with sea ice concentration nearing 100%. This suggests that these fluxes were dominated by the freezing process, whereby CO2 gas is expelled into the brine channels, where it can then exchange with both the water below and the air above (Nomura et al., 2006). The mean flux during the initial freeze-up of early fall was  $0.1 \pm 3.8$  mmol m<sup>-2</sup> d<sup>-1</sup>. During late fall the mean was  $1.1 \pm 1.5$  mmol m<sup>-2</sup> d<sup>-1</sup>. However, because the quality-controlled data are not a perfectly continuous record (due to data gaps), in order to gain a measure of the seasonal flux we integrated the area under the local regression line (Fig. 4) and divided by time. For the entire fall period this gave a flux of 0.38 g-C m<sup>-2</sup>. When excluding the two weeks of downward flux in October this value rose to 0.73 g-C m<sup>-2</sup>.

While the freeze-up period appears to have distinct parts separating air-water exchange from freezing-related flux, it is expected that both processes are occurring throughout. If so, the fluxes are simultaneously acting in opposite directions, acting to reduce the magnitude of the total flux (Fig. 10).

Figure 10. Theoretical diagram representing the competing downward air-sea fluxes with upward freeze-related component of the flux during marginal sea ice conditions. The seasonal freeze-only component of the flux can be calculated by integrating the area under its curve (hatched area). The designation of "solid ice" refers to the moment during winter when low sea ice temperatures render the ice matrix impermeable (Gosink et al., 1976).

To isolate the flux due to freezing in our dataset we estimated the flux through the open water areas using

$$F_{CO2} = f F_{BULK}, \tag{3}$$

where f is fraction of open water and  $F_{BULK}$  is  $CO_2$  flux calculated using the bulk formula (Eq. (1)). This value was then subtracted from the measured fluxes to obtain a freeze-only flux estimate. Because we did not measure  $pCO_{2w}$  we used a constant value of -49  $\mu$ atm in the calculation of the bulk flux, which was the  $pCO_{2w}$  estimate based on  $F_{CO_2}$  for October 14,

Deleted: et al.

• Deleted: et al.

under open water conditions just prior to freeze-up. The assumption that  $pCO_{2w}$  remains constant after the onset of freezing is based on there being minimal biological activity (Yendamuri et al., 2024), minimal water temperature changes to influence  $pCO_{2w}$  during this season, and mixed layer depths approaching the sea floor. This assumption is supported by a yearlong dataset of under-ice  $pCO_{2w}$  measured from an autonomous, underwater sensor platform in nearby Cambridge Bay, which showed only minor variations in  $pCO_{2w}$  after the onset of freezing (Duke et al., 2021).

Deleted: et al.

The bulk flux estimate suggests that without the influence of freezing the measured flux would have been consistently downward or zero (Fig. 11). Subtracting the bulk flux from the measured flux we get an estimate of the freeze-only flux. Diurnal variations in both wind speed and freeze-thaw during this freeze-up period complicate the assessment. This can be seen by the large variance in F<sub>CO2</sub> over short timescales in Fig. 11. However, by smoothing diurnal variations we observed that outgassing from freeze-only flux starts at the onset of freezing and continues through the fall season (Fig. 11 – green line).

Integrating the area under this curve we estimate a freeze-only flux of 1.07 g-C m<sup>-2</sup> for the season.

Figure 11. Shows the  $F_{CO2}$  during the freeze-up period (10/14 - 12/31) as points colored by sea ice concentration. The black curve represents a locally-weighted least-squares regression of measured  $F_{CO2}$  fit using a quadratic polynomial. The purple curve represents the same locally-weighted regression, but for the calculated bulk  $F_{CO2}$ . The green line represents the freeze-only component of the flux, calculated by subtracting the bulk flux from the measured  $F_{CO2}$ .

#### 3.5.2 Fall Discussion

Previous measurements of freezing-related outgassing from the initial formation of sea ice have been limited to laboratory studies. Laboratory tank experiments have found  $F_{CO2}$  over forming sea ice ranging from 0 to 1.0 mmol  $m^{-2}$   $d^{-1}$  (Nomura et al., 2006) and -0.4 to 0.75 mmol  $m^{-2}$   $d^{-1}$  (Kotovitch et al., 2016). Previous field studies have measured  $F_{CO2}$  over young sea ice soon after it formed and have found slightly larger (though still small) upward fluxes. Nomura (2018) measured  $F_{CO2}$  of  $3.7 \pm 2.0$  mmol  $m^{-2}$   $d^{-1}$  for young ice and  $0.7 \pm 0.7$  mmol  $m^{-2}$   $d^{-1}$  for older ice. These fluxes are the same order of magnitude

Deleted: et al.

as other chamber-based measurements over land fast ice like Nomura et al. (2013) and Delille et al. (2014), whose measurements over Antarctic pack ice showed a temperature dependence (i.e., T<−8°C → no flux; −8°C< T<−6°C → 1.9 mmol m<sup>-2</sup> d<sup>-1</sup>).

Our measurements span the range of these different ice regimes, and importantly include the period of initial ice formation.

During the week when SIC first reached 100% occurred (11/01 – 11/08) the mean measured F<sub>CO2</sub> was at a fall maximum at 2.6 ± 3.6 mmol m<sup>-2</sup> d<sup>-1</sup>. This <u>outgassing</u> agrees well with previous measurements over young ice, but is roughly a factor of 3 higher than F<sub>CO2</sub> measured by previous tank experiments. This may indicate the effect that wind has on increasing F<sub>CO2</sub>, a process which is absent from tank experiments. Additionally, this higher magnitude flux is seen in the freeze-only estimate of F<sub>CO2</sub>, which peaked for a month (10/22 – 11/22) at 1.7 ± 0.1 mmol m<sup>-2</sup> d<sup>-1</sup>. This peak period includes earlier periods of ice formation (i.e., before ice concentration reached 100%), meaning that the freeze-only portion of the flux was positive and of a similarly high magnitude, but was competing with downward air-sea F<sub>CO2</sub>.

Outgassing over the entire late fall period was lower, with a mean  $\frac{\Gamma_{CO2}}{\Gamma_{CO2}}$  of  $1.1 \pm 1.5$  mmol m<sup>-2</sup> d<sup>-1</sup>. Additionally, we found a seasonal/temperature trend, with fluxes decreasing from their highest magnitudes  $(2.6 \pm 3.6 \text{ mmol m}^{-2} \text{ d}^{-1})$  during the first week of November towards their lowest flux magnitudes  $(0.5 \pm 1.5 \text{ mmol m}^{-2} \text{ d}^{-1})$  during the last two weeks of December, when temperatures were colder and the ice was thicker. This fits previous findings that gas migration is more effective in warmer sea ice compared with colder sea ice, where the formation of brine is significantly reduced (Gosink et al., 1976; Delille et al., 2014). In practical terms this means that full, solid, cold ice cover acts as a barrier to gas exchange.

To put the freezing-related fluxes from this study into context we estimated an Arctic-wide flux from freezing. The area of Arctic first-year sea ice was estimated to be 9.4 million km², calculated as the average annual range of sea ice area over a five-year period from 2014 – 2018, based on the NSIDC monthly sea ice area for the northern hemisphere (Fetterer et al., 2017). Using the cumulative flux for the entire fall season at our site to extrapolate, we estimate the total Arctic CO2 outgassing from freezing for 2017 (10/14 – 12/31) was 6.8 Tg-C. If we use our freeze-only estimate (which removes the influence of downward air-sea gas exchange) that increases to 9.9 Tg-C. Bates and Mathis (2009) estimated an annual Arctic Ocean CO2 exchange of –66 to –199 Tg-C yr<sup>-1</sup>, a net sink. Our estimate for outgassing from the freeze-up period represents a counterbalance equivalent to 3.5 to 10% of this total Arctic sink, or 5 to 15% if we use our estimate for the 'freeze-only' component of the measured flux. While this is a rough estimate, it suggests that outgassing from freezing represents a small, but non-negligible portion of annual flux, which is not typically considered in Arctic CO2 budgets.

Another aspect of freeze-up that we were able to address is the previous hypothesis that gas exchange is enhanced in the presence of forming sea ice (Anderson et al., 2004; Else et al., 2011). To do so, we assumed that  $\Delta p CO_2$  remained constant

Deleted: Fco2

Deleted: et al.

Deleted: et al.

during the marginal ice conditions at the beginning of the freeze-up period (i.e., early fall). As described above, this assumption is rooted in evidence for minimal biological activity or temperature changes. We then set measured  $F_{CO2}$  equal to Eq. (1) to estimate gas transfer velocity normalized to a Schmidt number of 660 ( $k_{660}$ ) and weighted it to the fraction of open water.

Figure 12. Shows gas transfer velocity versus wind speed for the period of 10/14 to 10/28 when the region exhibited marginal ice conditions (e.g., 0 < SIC < 100%). These  $k_{660}$  values are weighted to the fraction of open water for comparability to Wanninkhof (2014).

The magnitudes of k<sub>660</sub> during the freeze-up period with marginal ice conditions (10/14 – 10/28) stayed relatively close to open water relationships of k<sub>660</sub> and 10-m wind speed (Fig. 12). The scatter in the figure was most likely due to the fact that ΔpCO<sub>2</sub> was held constant at –49 μatm, which was unlikely to have been rigidly the case through this period. Without ΔpCO<sub>2</sub> measurements we have no way to determine whether fluxes were enhanced in minor ways (e.g., say 20%). But our data does contradict previous findings of large enhancements (e.g., orders of magnitude) to gas exchange in the vicinity of sea ice. Such a scenario would have been characterized by our k<sub>660</sub> far surpassing open water parameterizations.

## 3.6 Winter

710

We do not have a continuous record of overwinter  $F_{CO2}$  because power constraints halted data collection from January to April. However, we can gain information about fluxes during this period from April and May measurements, when there was full ice cover and air temperature remained below  $0^{\circ}$ C. During this period the mean  $F_{CO2}$  is low at  $-0.04 \pm 0.40$  mmol m<sup>-2</sup> d<sup>-1</sup>. Because air temperatures during the winter are typically below  $-20^{\circ}$ C (i.e., below the temperature at which sea ice matrix becomes impermeable [Gosink 1976]), we expect that the winter mean flux does not exceed this pre-spring mean flux. If true, that would

put an upper boundary on the cumulative winter flux at -0.1 g-C m<sup>-2</sup>, or 1% of the annual flux. Because it is speculative, we omitted this value from the annual sum of seasonal fluxes in Table 2.

#### 4 Process Summary

Major variables influencing F<sub>CO2</sub> in this region are temperature, <u>salinity</u> melt, ice formation, mixing, and biological activity.

Figure 13 shows the relative timing and peak influence of these variables as reflected in the flux measurements from 2017. Over the winter there appears to be very little flux. In early spring the processes that appear to influence the fluxes are melting, freezing, and primary production. Both ice melt and photosynthesis cause *p*CO<sub>2w</sub> to decrease, which results in downward F<sub>CO2</sub>. The influence of melt only lasts while sea ice is present, but the drawdown due to photosynthetic activity could potentially last into the later stages of spring (though the magnitude of its influence is expected to be small due to the nitrogen-limited seawater in this region [Williams et al., 2025]).

As the ice starts to break up the influence of increasing SST provides a positive forcing in opposition to the melt and biological activity. Changes in SST are prominent through the open water summer season, with increasing SST in early summer leading to outgassing, while decreasing SST in mid and late summer providing a negative forcing on the flux. Though weaker than the SST effect, salinity trends were also relevant to the thermodynamic forcing. In early and mid summer,  $S_{SW}$  decreased, causing a negative forcing on  $pCO_{2w}$ . In late summer  $S_{SW}$  began to increase, leading to a positive forcing on  $pCO_{2w}$ . Mid and late summer are also characterized by an increasing mixed layer depth, which may result in high  $pCO_{2w}$  water from lower depths mixing to the surface, providing a positive forcing on the flux in opposition to the forcing from decreasing SST. In fall, the mixed layer depth approaches the sea floor, biological activity (both respiration and photosynthetic) has mostly ceased, and the SST can drop no further. Salinity does still increase at this point, but across the early fall period its contribution towards increasing  $pCO_{2w}$  was modest ( $+10 \mu$ atm). This appears to make the process of freezing-related outgassing the most prominent influence on the flux during this time.

Deleted: et al.

Figure 13. Schematic diagram showing the direction and timing of the environmental processes influencing Fco2 in the spring. summer, and fall seasons, based on fluxes from 2017. The peaks represent the estimated time of maximum influence for each individual process (magnitudes are arbitrary).

The direction of fluxes that we measured across the annual cycle were in general agreement with  $\Delta p$ CO<sub>2</sub> gradients measured by Sims et al. (2023) within a ~100 km radius of the flux station. Sims et al. (2023) did note substantial spatial variability, which makes it difficult to confidently extrapolate the net annual flux over a larger area. However, an estimate of k calculated using tower F<sub>CO2</sub> and ship-based pCO<sub>2w</sub> measurements of Sims et al. (2023) during temporally-aligned courses past the island showed good agreement with existing open-water k parameterizations, providing evidence the capability of the tower-based F<sub>CO2</sub> for estimating pCO<sub>2w</sub> (Butterworth and Else, 2018).

While other processes (e.g., stream discharge, tidal cycle, etc.) are expected to be relevant at various points throughout the year, they are expected to be more minor influences on Fco2 relative to these main processes. The tidal cycle was investigated for a relationship with F<sub>CO2</sub> and no correlation was found. Future research from this site may be able to highlight the magnitude of individual processes with greater precision. Due to its relevance to the F<sub>CO2</sub> cycle, direct measurements of pCO<sub>2w</sub> were collected at the site during subsequent years. These were made possible by the installation of a mobile power station/research lab (with sleeping quarters), installed on the island in 2018. These measurements will be incorporated into future research investigating CO<sub>2</sub> gas transfer velocity continuously through the annual cycle.

#### 5 Conclusions

The goal of this study was to determine the biogeophysical factors influencing F<sub>CO2</sub> in an Arctic marine environment through an entire annual cycle. An eddy covariance system enabled the collection of flux observations during periods which have been traditionally difficult to capture by methods with limited temporal scope (e.g., chamber measurements, ship-based eddy covariance). At this site we found that the annual net CO<sub>2</sub> flux was small at only -0.3 g-C m<sup>-2</sup>. However, this annual flux was composed of larger counteracting positive and negative fluxes in the different seasons.

In the spring seasons the measurements provided in situ evidence for CO<sub>2</sub> uptake during melt pond and ice break-up. In the summer seasons, we found that SST played a major role influencing FCO2. The collection of CO2 flux measurements during the fall freeze-up period represented a unique aspect of this dataset. As far as we know, this was the first field campaign to collect eddy covariance CO2 flux measurements over newly forming sea ice. The measurements provided in situ evidence for theoretical and laboratory findings that ice formation leads to positive (upward) CO<sub>2</sub> flux. The measurements suggest that air-

**Deleted: Figure 13.** Schematic diagram showing the direction and timing of the environmental processes influencing Fc02 in the spring, sum and fall seasons, based on fluxes from 2017. The peaks represent the estimated time of maximum influence for each individual process (magnitudes are arbitrary).

Deleted: FC

Deleted: The direction of fluxes that we measured were in general agreement with ΔpCO2 gradients measured by Sims et al. (2023) within a ~100 km radius of the flux station. However, Sims et al. (2023) did note substantial spatial variability, which makes it difficult to confidently extrapolate the net annual flux over a larger

ice fluxes of CO<sub>2</sub> during the freezing process are not negligible, as some studies have suggested, and may <u>produce a counterbalancing outgassing equivalent to 5 - 15% of the annual Arctic CO<sub>2</sub> sink. Therefore, we recommend their inclusion in future modeling of polar marine carbon budgets.</u>

Deleted: account for

Deleted: budget

The collection of data over two seasons also provided some preliminary insights into interannual variability. The timing of the start of spring melt appeared to play a role in the maximum CO<sub>2</sub> uptake reached during the summer (i.e., earlier melt leading to greater uptake). This is consistent with high observed interannual variability of Δ*p*CO<sub>2</sub> in the region, which Sims et al. (2023) found was related to timing of sea ice break-up. The timing of mixed layer deepening (i.e., earlier melt leading to later deepening), also appeared to play an important role through the delivery of high-*p*CO<sub>2</sub>w water from depth. It may help explain why a late melt year like 2018 did not transition to a CO<sub>2</sub> sink at the beginning of September, while an early melt year like 2017 did. However, with F<sub>CO2</sub> measurements in 2018 terminating on 09/15 (due to instrument failure) we cannot dismiss the possibility that a CO<sub>2</sub> sink developed later in fall 2018 as water temperatures continued to decrease. Because many previous studies of Arctic CO<sub>2</sub> flux have relied upon observations and measurements taken during the summer season, the prevalence and importance of this fall sink to the Arctic carbon budget has, to this point, not received attention. This is a potentially important process and one which may become more prevalent as the Arctic further warms.

This work shows that with appropriate system design  $F_{CO2}$  measurements can be made continuously in harsh Arctic conditions and that those measurements can be effectively deployed to address a range of potential research questions. Additionally, such  $F_{CO2}$  measurements promise to be highly useful for research on biogeochemical processes in the Arctic marine environment, particularly if they can be extended to other sites with different ice, ocean, and atmospheric conditions.

# Data availability

The data used for this research have been published on the Zenodo data repository (Else and Butterworth 2025). They can be found at the following link: <a href="https://doi.org/10.5281/zenodo.15191010">https://doi.org/10.5281/zenodo.15191010</a>

# 825 Author contribution

BJB and BGTE designed and installed the flux system. BGTE secured the grant funding for the research activities, and organized field logistics. BJB processed and analyzed the flux data. BJB prepared the manuscript with contributions from BGTE, KAB, CJM, WJW, LMR, and GdB.

#### Competing interests

The authors declare that they have no conflict of interest.

#### Acknowledgements

- We wish to thank the students and technicians who helped install and maintain the eddy covariance tower; in particular

  Shawn Marriott, Patrick Duke, Angulalik Pederson, Jasmine Tiktalek, Laura Dalman, and Vishnu Nandan. We would also like to thank Yuanxu Dong and one anonymous reviewer for their constructive reviews. The deployment of this tower would not have been possible without the excellent logistical support provided by the Arctic Research Foundation, Polar Knowledge Canada, and the Polar Continental Shelf Program. Financial support was provided by the Natural Sciences and Engineering Research Council of Canada (NSERC: Discovery grant number RGPIN-2015-04780), the Marine
- Environmental Observation Prediction and Response (MEOPAR) Network of Centres of Excellence, Polar Knowledge Canada, the Canada Foundation for Innovation John R. Evans Leaders Fund, the Nunavut Arctic College, Irving Shipbuilding Inc., and the University of Calgary. We would also like to thank the Ekaluktutiak Hunters & Trappers Organization for the expert assistance provided by their guides. This paper is a contribution to the SCOR Working Group 152 Measuring Essential Climate Variables in Sea Ice (ECV-Ice). Additional support was provided by the US Department of Energy Atmospheric System Research program, through project number DE-SC0013306.

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
