# Peer review of "Annual carbon dioxide flux over seasonal sea ice in the Canadian Arctic"

_EGUsphere, 2025_

## Referee Comment (RC2)

The interactions among air, water, and ice have long been recognized as critical for accurately estimating gas fluxes in polar oceans. However, measuring $CO_2$ fluxes in natural sea-ice-covered regions remains extremely challenging, particularly due to the logistical difficulties of conducting long-term eddy covariance (EC) observations in such environments. This study presents 17-month EC measurements of air-sea $CO_2$ fluxes in a coastal, ice-covered setting, which is a significant contribution to the field. The dataset clearly captures the temporal variability of $CO_2$ fluxes across multiple timescales. Notably, the identification of $CO_2$ outgassing associated with ice formation is a novel and important finding that could have substantial implications for refining estimates of the polar ocean carbon sink.

The manuscript is well written, the results are clearly presented, and the conclusions are scientifically sound. I believe the paper is suitable for publication after addressing the following minor comments.

PS, the comments from the other reviewer is referred.

Minor comments,

Lines 45–50: I suggest including the recent study by Prytherch and Yelland (2021), which is a dedicated investigation of the influence of sea ice on $CO_2$ exchange: https://doi.org/10.1029/2020GB006633 . While it is cited in line 89, it appears to be missing from the bibliography.

Line 125: It appears that a LI-COR 7500 sensor is installed on the tower, but LICOR7200 does not appear. I see in you 2018 paper, the LI-COR 7500 was used to measure water vapor and $CO_2$ was measured by LICOR7200? Could you clarify here?

Line 180: It would be helpful to provide more explanation of the Kice term, specifically, how it was derived or constrained.

Figure 3: Could you include information about wind direction? It seems that some flux data may be missing due to winds coming from the direction of the island?

Figure 4: The high-frequency time series is 6-hour averaged. Readers may be interested in the extent to which the observed variability is influenced by EC uncertainty. Could you provide at least a simple estimate of the uncertainty magnitude, include that value in the figure caption, and briefly discuss it in the main text?

Figure 5: I agree with the other reviewer that the possibility of $pCO_2$ice being negative should be explained. I suspect the derived values may be sensitive to the estimation of Kice. While some discussion is included later in the manuscript, it would be helpful to provide an earlier explanation, perhaps around line 185.

Line 275: You mention that camera images were collected, but none are shown in the paper, which is a shame. Would it be possible to include several representative images from different stages of the observation period? These could be placed alongside Table 1 or included in the supplementary material.

Line 333: For your reference, we have conducted a related study using eddy covariance and pCO2w measurements in an ice melt region, which indicates substantial $CO_2$ uptake: Dong et al. (2021), Geophysical Research Letters, https://doi.org/10.1029/2021GL095266 .

Line 370: The first sentence in this paragraph reads awkwardly to me. Please consider rephrasing for improved clarity and flow.

Line 533: The square brackets around the reference should be removed to maintain consistency with the formatting style.

Line 569: The abbreviation "EC" appears here without prior definition.

Conclusions: The comparison with Sims et al. (2023) is valuable, but might be more impactful if introduced earlier in the discussion section. As currently presented, it reads more like a discussion point than a concluding remark.

Final suggestion: It may strengthen the conclusions if you emphasize that concurrent measurements of $pCO_2$w would provide more robust support for some of the interpretations presented in this study.

-Yuanxu Dong

---

## Author Response (AR1)

**Response to Reviewer 1**

Reviewer comments in black.
Author responses in blue.
New manuscript text in red.

**Reviewer 1 Comments**

This paper shows the continuous measurements of CO2 flux collected from a 10-m eddy covariance tower in a coastal-marine environment in the Canadian Arctic Archipelago over the course of a 17-month period. The extended length of data collection resulted in a unique dataset that includes measurements from two spring melt and summer seasons and one 20 autumn freeze-up. Generally, this paper is written well and conclusion is clear. However, for explanation of the pCO2 sw calculated based on the FCO2, K, and pCO2 air are unrealistic values (e.g. zero). Therefore, author should explain and adjust for the appropriate explanation.

We would like to thank Reviewer 1 for their insightful comments. We have addressed the reasoning behind the large swings in pCO2w below, in response to specific comments. We also found their suggestion to include salinity effects to be an important addition to the manuscript. We reworked the summer analysis to better highlight the dominance of the thermodynamic processes on  $F_{CO2}$  during that season.

Comments are indicated in the following.

Line 21: "air-sea" should "air-sea" (wider minus) throughout the text.

While the case for choosing the en dash to connect "air" and "sea" makes perfect sense (i.e., read as "air to sea exchange"), we prefer the hyphen to connect "air" and "sea". Grammatically, it has a subtle difference from the en dash version, namely to create a compound adjective to describe "exchange", "fluxes", etc., which is still within the bounds of acceptable usage. Moreover, this has been the convention used by many respected scientists for decades.

Line 21: Author indicated that sea ice is barrier. However, in lines 28–30, outgassing occurs during freezing sea ice. Author wanted to indicate zero after freezing. Therefore, to avoid misunderstanding, author should write correctly.

To disambiguate the flux barrier of thick winter ice from the outgassing that occurs at the beginning of the freezing process we made the following changes:

- Changed "full sea ice cover" to "thick ice"
- Changed "winter field campaign" to "autumn period of the field campaign"

Line 29: For positive flux values, it is nice if author will add "+" throughout the text.

We prefer not to add "+" for positive values. It is accepted as standard that values not preceded by a minus sign are positive. We think the "+" looks clunky, especially followed by "±" standard deviation. That said, we understand that directionality was particularly important to this work. For most flux values presented we had identified the direction in text form (e.g., "uptake", "outgassing", "sink", "source") within the paragraph. We checked the manuscript an added these qualifiers (where they were absent) to help make directionality more clear.

Lines 32–33: "CO2 outgassing from the freezing period to be 5 to 15% of the magnitude of the estimated Arctic CO2 sink". If same direction of flux, we can say the percentage with respect to total. However, it is different direction (positive and negative). Can we say 5 to 15%?

Yes, we don't see why not. It states that it is the opposite direction. The purpose here was only to identify the relative magnitude of this generally unaccounted-for process. To highlight that it certainly warrants some attention. We've modified the sentence to include "a counterbalance equivalent to" in front of "5 to 15%" to highlight the opposing direction more clearly.

Line 55: CaCO3 salts. We do not need "salts".

We removed "salts"

Line 156: Author should add the detection limit, standard deviation, and accuracy of FCO2.

**Yes, that is relevant information. We have added the following text:**

As the product of measurements from different instruments, the accuracy of the  $F_{\rm CO2}$  measurement is challenging to quantify without an independent validation, which was not performed. The LI-7200 has a measurement accuracy of  $\pm 1\%$  with an RMS noise of 0.11 ppm at 10 Hz, while the vertical wind speed of the CSAT3 is accurate within  $\pm 0.04$  m s-1 with an RMS noise of 0.0005 m s-1. While the noise can occasionally be larger than the true environmental fluctuations, it has been found to minimally influence the calculated  $F_{\rm CO2}$  because the noise from the separate instruments is uncorrelated and therefore filtered out by the flux calculation (Miller et al., 2010).

An investigation of  $F_{CO2}$  measurement uncertainties from ships indicated a detection limit for a dried, closed-path eddy covariance system of roughly  $|\Delta p CO_2| > 35$  µatm for the mean wind speed observed in this study (Blomquist et al., 2014). The  $\Delta p CO_2$  in the region often exceeds this value (Duke et al., 2021; Sims et al., 2023). Additionally, we expect some reduction in the detection limit (i.e., increased sensitivity) for this study compared to ship-based studies, because the measurements were from a stationary tower. Therefore, the observations avoid some common sources of uncertainty experienced from moving platforms, such as the needed for a complex wind vector motion correction and tilt effects that degrade the performance of the LI-7200 (Miller et al., 2010; Vandemark et al., 2023).

While we cannot perform a direct assessment of  $F_{\rm CO2}$  uncertainty, we can estimate the order of magnitude of the uncertainty by assessing the variation in  $F_{\rm CO2}$  measurements during periods expected to have stable fluxes. Here we do that by calculating the standard deviation for 6-hour intervals during periods of full ice cover, when diurnal variations in  $F_{\rm CO2}$  were expected to be minimal. The standard deviation across these winter periods had a mean of  $\pm 1.02$  mmol m-2 d-1 and a median of  $\pm 0.75$  mmol m-2 d-1. Spring and summer seasons were excluded from the estimate because standard deviation measured during those periods was expected to be a combination of measurement uncertainty and actual diurnal  $F_{\rm CO2}$  trends.

**Added the following text to the Fig. 4 caption:**

Uncertainty in the  $F_{CO2}$  measurement was quantified by calculating the standard deviation from each 6-hour average (comprised of eighteen 20-minute flux intervals) during periods of full ice cover, when diurnal  $F_{CO2}$  variations were minimal. The standard deviations across these winter periods had a mean of  $\pm 1.02$  mmol m-2 d-1 and a median of  $\pm 0.75$  mmol m-2 d-1.

**Added the following reference to the bibliography:**

Vandemark, D., Emond, M., Miller, S. D., Shellito, S., Bogoev, I., and Covert, J. M.: A CO2 and H2O Gas Analyzer with Reduced Error due to Platform Motion. J Atmos Ocean Technol, 40, 845–854. doi: 10.1175/JTECH-D-22-0131.1, 2023

Line 176: CO2 in the brine?

Added: (including brine)

Line 197: Section of "Results and Discussion" will be divided into "Results" section and "Discussion" section.

We deliberately organized the manuscript in an unusual way. The reason for a "Results and Discussion" section was to enable the results from each season to be followed by a discussion of that season (i.e., "Spring Result" followed by "Spring Discussion", etc). In an early version of the manuscript, we had the more traditional format, but found that the varied processes occurring in each season made holding all the results in memory too difficult. We found that the individual seasons basically acted as their own small studies and therefore were more easily discussed immediately following their corresponding results.

Line 199: Freezing point of seawater?

Added: of seawater

Line 341: Sea ice melt water affects low pCO2 sw due to dilution effect in lead water etc. Author will indicate the potential effect of pCO2 dilution effect by the melt water supply based on salinity data.

**We added the following text:**

The salinity mooring data was inspected to determine whether melt water dilution was observed. At 13 m depth there was a small decrease in  $S_{SW}$  (-0.2) over the late spring period. This would correspond to a small decrease in  $pCO_{2w}$  (-3  $\mu$ atm), a relatively small  $F_{CO2}$  forcing. However, because the water was stratified at this period (i.e.,  $SST > T_{13m}$ ), it is possible (and likely) that the change in  $S_{SW}$  at the surface was greater, resulting in a larger  $F_{CO2}$  forcing.

Line 370: It would be nice if author will compare with SST and pCO2 because pCO2 will change depend on temperature. Author can indicate that this pCO2 change can explain based on thermodynamic process or not.

We replaced Figure 9 with a new plot showing the pCO2w projected forward from the first day of summer using the temperature and salinity relationships of Takahashi et al. (1993) and Sarmiento and Gruber (2006). These are compared to pCO2w estimated using Eq. 1 with the measured FCO2. In both early and mid summer there is good agreement/tracking between both estimates. This suggests that thermodynamic processes are the most relevant during these summer seasons. We also added a salinity curve to Figure 13 to highlight its directional impact on the FCO2 across seasons. We added the following text to the manuscript:

**Added to Summer Results:**

We separated the summer season into three subseasons (early, mid, and late) corresponding to changes in environmental conditions. The early season (13 July 2017 – 13 Aug 2017) was from the beginning of open water until peak SST and showed increasing  $F_{CO2}$  (Fig. 8). The mid season (13 Aug 2017 – 1 Oct 2017) was from peak SST until the water profile became unstable and showed decreasing  $F_{CO2}$  (Fig. 8). The late season (1 Oct 2017 – 14 Oct 2017) was the period immediately preceding the onset of freezing in which the mixed layer deepened. We then investigated the role of thermodynamic processes on the observed seasonal  $F_{CO2}$  changes. Figure 9 shows a  $pCO_{2w}$  estimate derived from  $F_{CO2}$  using Eq. (1) and a  $pCO_{2w}$  projection calculated using established temperature and salinity relationships ( $\frac{1}{pCO_{2w}} \frac{\partial pCO_{2w}}{\partial SST} \approx 0.0423^{\circ}C^{-1}$  from

Takahashi et al. [1993];  $\frac{S_{SW}}{pCO_{2W}} \frac{\partial pCO_{2W}}{\partial S_{SW}} \approx 1$  from Sarmiento and Gruber [2006]). For the thermodynamic projection, the  $F_{CO2}$ -derived  $pCO_{2W}$  estimate for the first day of early summer was used as a starting  $pCO_{2W}$ , then projected forward for each flux interval through the end of summer using only the above SST and  $S_{SW}$  relationships. In both early and mid summer, the two  $pCO_{2W}$  estimates track well, indicating that changes in SST and  $S_{SW}$  are important drivers of  $F_{CO2}$  changes during these seasons. In late summer, the curves show greater divergence with the  $F_{CO2}$ -derived  $pCO_{2W}$  estimate showing larger values (over 25  $\mu$ atm greater) than the thermodynamic projection. While the  $F_{CO2}$  increased from its seasonal low during this late summer period (Fig. 4 & 8; due to the reduced wind speed [Fig. 3e]), the  $F_{CO2}$ -derived  $pCO_{2W}$  estimate continued to

drop in magnitude (Figs. 7 & 9). This was in opposition to the SST forcing, but coincided with deepening of the mixed layer and increased SSW values.

**Added to Summer Discussion:**

In summer, thermodynamic drivers appear to be the most important contributors to the direction and magnitude of FCO2. For most of the summer, the trend in FCO2 corresponds to the trend in SST. Both increase in early summer, both decrease in mid summer (Figs. 4 & 8). The mechanism causing this pattern is the direct positive relationship between SST and pCO2w (Takahashi et al., 1993). As SST increases, it causes  $pCO_{2w}$  to increase, which results in increased outgassing of  $CO_2$  to the atmosphere.  $S_{SW}$  also has a direct positive relationship with  $pCO_{2w}$  (Sarmiento and Gruber, 2006). In this instance, steady reductions in SSW over the course of the early and mid summer periods (28 down to 25) partially offsets the projected peak magnitude of  $pCO_{2w}$  by the SST effect alone. The projection of pCO2w using both SST and SSW effects tracks well with the FCO2-derived pCO2w estimate (Fig. 9). This suggests that SST and SSW are the main drivers of changes to FCO2 in the early and mid summer periods. The one period of the summer in which the thermodynamic pCO2w projection most noticeably diverges from the FCO2-derived pCO2w estimate is late summer (1 Oct 2017 – 14 Oct 2017). During this period the SST continues to drop, but SSW begins to increase (25 up to 27). This corresponds to a reduced (but still negative) slope to both the FCO2-derived pCO2w estimate and the thermodynamic projection. The cause of the increased SSW was the reversal of the temperature profile from stable to unstable (i.e., SST <  $T_{13m} < T_{22m} < T_{39m}$ ) resulting in greater upward mixing of higher salinity water from depth. While the similar trends in both the  $pCO_{2w}$  estimate and the thermodynamic projection suggest that SST and SSW are still important drivers of FCO2 during late summer, the higher magnitudes of the pCO2w estimate compared to the thermodynamic projection suggest an additional source of increased pCO2w. One possibility is that the increased mixing of water from depth during this late summer period may have, in addition to increasing SSW, brought CO2-rich waters to the surface, thus slightly offsetting some of the  $pCO_{2w}$  reductions expected by the thermodynamic processes alone.

As stated above, the pattern of  $F_{CO2}$  in 2018 was similar to 2017, with the exception of 2018 showing increasing positive fluxes and increasing  $pCO_{2w}$  in the first two weeks of September, running in opposition to the SST forcing. One explanation is that the lower SST during 2018 enabled mixed layer deepening a month earlier than the previous year, causing mixing to increase  $pCO_{2w}$  (e.g., due to  $S_{SW}$  and  $CO_2$  concentration effects) earlier in the season. Unfortunately, the mooring temperature and salinity data were not available during this period to confirm. However, an inspection of the flux cospectra during this period showed no reason to discount this upward trend on the grounds of flux measurement error.

**Added to Process Summary description of summer:**

Though weaker than the SST effect, salinity trends were also relevant to the thermodynamic forcing. In early and mid summer,  $S_{SW}$  decreased, causing a negative forcing on  $pCO_{2w}$ . In late summer  $S_{SW}$  began to increase, leading to a positive forcing on  $pCO_{2w}$ .

**Added to Process Summary description of fall:**

Salinity does still increase at this point, but across the early fall period its contribution towards increasing  $pCO_{2w}$  was modest (+10  $\mu$ atm).

**Added new Fig. 9 caption:**

Summer 3-day average time series of  $pCO_{2w}$  derived from  $F_{CO2}$  using Eq. (1) (black line) and  $pCO_{2w}$  projection calculated using temperature and salinity relationships (Takahashi et al., 1993; Sarmiento and Gruber, 2006; blue line). Shaded regions represent standard deviation.

**Added to bibliography:**

Sarmiento, J. L. and Gruber, N.: Ocean Biogeochemical Dynamics, Princeton University Press, https://doi.org/10.2307/j.ctt3fgxqx, 2006.

\*Also changed Takahashi et al. 2002 citations throughout the manuscript to Takahashi et al. 1993 citation because that was the original paper that published the T-pCO2 relationship, so therefore it is the more appropriate article to cite.

Line 399, "aqueous" does not need.

**Removed.**

Line 400; For biological process, author should use reference (biological paper showing about this area) to show author's explanation in the text.

In response to the Line 370 comment above, we reworked this section entirely. We no longer refer to biological activity in this section.

Line 415: How about the high pCO2 water mixing with surface water? Because this area is polynya and high current. Can Duke et al. (2021) (pCO2 data) support author's conclusion?

No, the pCO2 measurements from Duke et al. (2021) do not support the idea of high CO2 concentrations at depth. For some of the year, they showed similar magnitudes as the tower estimates, at other times they show pCO2w values far lower than those estimated using the tower measurements. It's not clear why this is the case. But that particular mooring was 35 km away. And the ship-based pCO2w measurements of Sims et al. (2023) showed that this region has a high degree of spatial variability in pCO2w. So, the values from that mooring are likely not comparable to our site.

Lines 504–505: If minus 9.9 Tg-C, we can say that 3.5 to 10 percent of this total Arctic sink. However, 9.9 is positive. Therefore, can we say 3.5 to 10 percent of this total Arctic sink?

It now reads "Our estimate for outgassing from the freeze-up period represents a counterbalance equivalent to 3.5 to 10% of this total Arctic sink".

Line 572: Author can indicate Sims et al. (2023) in the pCO2 discussion which will help author's assumption of relationships between flux and pCO2.

Moved Sims et al. (2023) discussion from Conclusions to Process Summary section, and added additional text. It now reads:

"The direction of fluxes that we measured across the annual cycle were in general agreement with  $\Delta p CO_2$  gradients measured by Sims et al. (2023) within a ~100 km radius of the flux station. Sims et al. (2023) did note substantial spatial variability, which makes it difficult to confidently extrapolate the net annual flux over a larger area. However, an estimate of k calculated using tower  $F_{CO2}$  and ship-based  $p CO_{2w}$  measurements of Sims et al. (2023) during temporally-aligned courses past the island showed good agreement with existing open-water k parameterizations, providing evidence the capability of the tower-based  $F_{CO2}$  for estimating  $p CO_{2w}$  (Butterworth and Else, 2018)."

Lines 581–582: Same comments as lines 504–505.

Changed: "may account for 5-15% of the annual Arctic CO2 budget" to "may produce a counterbalancing outgassing equivalent to 5-15% of the annual Arctic CO2 sink"

Line 567: Only physical factors?

Changed "physical" to "biogeophysical"

Table 1: Flux means CO2 flux?

Yes. We changed "Flux direction" to " $F_{CO2}$  direction" in table and caption And we changed "delta in flux" to "delta in  $F_{CO2}$ " in table and caption In Table 2 we changed "Total Flux" to "Total  $F_{CO2}$ " in table.

Table 1: It is unknown how to decide the date for seasonal transition.

Added the following text to the end of the caption for Table 1:

Seasonal cutoff dates were determined by transition to different defining processes, as identified by in situ observations from site visits and camera images.

Table 2: "Note that in this instance only the use of terms spring, summer, and fall are defined based on the zero crossings of the local regression curve from Figure 3 (and therefore straddle the seasonal demarcations defined in Table 1).". I cannot understand this. Could author explain detail?

We were referencing the incorrect figure. It should have been Figure 4. We have fixed the error.

The idea here is that throughout the annual cycle the flux direction oscillates between negative and positive. The point of Table 2 is to show the relative magnitude of these different periods of uptake and outgassing. We do this by integrating the area under the curve in Figure 4. It is not a precise measure of seasonal uptake and/or outgassing, but it does give a good estimate of the relative magnitude. We used the terms "spring uptake", "summer outgassing", "summer uptake", and "fall outgassing" to categorize these periods. These do not precisely line up with the seasonal demarcations used throughout the paper because those seasonal demarcations straddle the transitions from uptake to outgassing, and from outgassing to uptake. If we used the seasonal demarcations used for the other analyses, we would end up integrating positive and negative values and therefore would reduce the measured magnitude of those periods of uptake and outgassing. To improve the readability, we have modified the relevant caption sentences.

**Changed from the original:**

"Note that in this instance only the use of terms spring, summer, and fall are defined based on the zero crossings of the local regression curve from Figure 3 (and therefore straddle the seasonal demarcations defined in Table 1). The cumulative fluxes in this table were calculated by integrating the area under this curve."

**To:**

"The cumulative fluxes were calculated by integrating the area under the local regression curve from Figure 4 between the zero crossings separating periods of uptake from periods of outgassing. In this instance only, the use of terms "Spring", "Summer", and "Fall" are defined based on these zero crossings, identified in the "Dates" column of the table. Note that they are not precisely aligned with seasonal demarcations defined in Table 1 (which are used in all subsequent analyses). This was done to avoid integrating using seasonal demarcations that straddled positive and negative flux transitions."

Figure 1: It is nice if author will show the island shape and position of tower in the island.

Added an inset to the map to show the island shape and position of the tower. Added the following text to the caption:

"Circular inset shows the shape of Qikirtaarjuk Island, with the red dot indicating the location of the flux tower."

Figure 4: Author should add 10 and –10 for the vertical axis.

Done.

Figure 4: Why FCO2 deviated widely during late spring to early fall as compared to winter and late fall?

The larger magnitude fluxes during late spring to early fall are mainly due to the fact this period has open water. The processes described in the paper (e.g., biological processes, SST changes, salinity changes, water mixing) are responsible for the flux direction oscillations during this period. Whereas the winter and late fall are ice covered, which leads to lower magnitude fluxes due to the ice acting as a barrier to exchange (except for some expulsion of CO2 from ice during the freezing process in late fall).

Figure 5: I cannot imagine that pCO2 become from 600 to zero within 9 hours. What is the mechanism driving such a big change? I expect that eq1 is not fit. How about the ice temperature change during 9 hours?

Yes, we called to the wrong equation. It should have been Equation 2. This has been fixed.

Our interpretation of pCO2 swing from 600 to zero within the course of a day is that melt ponds rapidly form, have pCO2 near zero when they melt (because CO2 was expelled during the freezing process), and then uptake CO2 after forming, and finally expel the CO2 they've absorbed as they refreeze at night. As referenced in the Spring Discussion section pCO2 of melt ponds have been measured an found to be near zero (Geilfus et al., 2015), and that they rapidly equilibrate after melting. Interestingly, the switch from uptake to outgassing indicates a sign direction in dpCO2, which at first is puzzling because if the melt ponds are undersaturated the equilibration after their melt should only bring them back to ambient air pCO2 values (at maximum). This wouldn't support the outgassing observed. The expulsion of CO2 from water as it freezes is the relevant process, but it's aided by the fact that it is only the surface ice which exchanges with the air. Supersaturation appears to occur near the surface. This is in line with Kotovich et al. (2016) who found outgassing due to super saturation occurring in the top 5 cm of newly formed ice, while the water below remained under-saturated. We have added this information to the text...

**Changed from the original:**

"Additionally, the estimated range of pCO2ice in this study matches that measured by Delille et al. (2014) in Antarctic pack ice (roughly  $50-900 \mu atm$ ). While that study represents seasonal changes in pCO2ice, it shows that pCO2ice of these magnitudes ( $0-600 \mu atm$ ) are plausible."

**To:**

"The low  $pCO_2$  of melt ponds are expected to immediately begin to equilibrate toward atmospheric values (Geilfus et al. 2015). However, the diurnal change in flux direction from uptake to outgassing indicates that  $pCO_{2ice}$  rose above atmospheric values. This suggests that the  $CO_2$  gas expelled during freezing accumulated in a thin, supersaturated layer near the surface. This is in line with the laboratory experiment of Kotovich et al. (2016), who also observed outgassing during freezing due to supersaturation in the top 5 cm of ice, while the underlying

water remained undersaturated with respect to the atmosphere. The large range of  $pCO_{2ice}$  in this study has some analogies in the literature. This includes the range measured by Delille et al. (2014) in Antarctic pack ice (roughly  $50-900~\mu atm$ ) and the range observed by Geilfus et al. (2015) in Arctic springtime ice ( $36-380~\mu atm$ ). While these studies represent daytime-only  $pCO_{2ice}$  measurements over longer time frames (seasonal and sub-week, respectively), they show that  $pCO_{2ice}$  of these magnitudes ( $0-600~\mu atm$ ) are plausible."

Unfortunately, we do not have a measure of ice temperature to report.

Figure 9: Author will check the relationship between SST and pCO2, also SSS to understand the direct relationships between pCO2 and environmental factors.

See response to comment about Line 370 above.

**Response to Reviewer 2**

Reviewer comments in black. Author responses in blue. New manuscript text in red.

**Reviewer 2 Comments**

The interactions among air, water, and ice have long been recognized as critical for accurately estimating gas fluxes in polar oceans. However, measuring CO2 fluxes in natural sea-ice-covered regions remains extremely challenging, particularly due to the logistical difficulties of conducting long-term eddy covariance (EC) observations in such environments. This study presents 17-month EC measurements of air-sea CO2 fluxes in a coastal, ice-covered setting, which is a significant contribution to the field. The dataset clearly captures the temporal variability of CO2 fluxes across multiple timescales. Notably, the identification of CO2 outgassing associated with ice formation is a novel and important finding that could have substantial implications for refining estimates of the polar ocean carbon sink.

The manuscript is well written, the results are clearly presented, and the conclusions are scientifically sound. I believe the paper is suitable for publication after addressing the following minor comments.

PS, the comments from the other reviewer is referred.

We would like to thank Dr. Dong for his comments. We have incorporated his suggestions to further develop the description of our Kice analysis and add an  $F_{CO2}$  uncertainty analysis. We also appreciate the minor comments pointing out things we missed and the suggestions to improve the readability of the manuscript.

**Minor comments,**

Lines 45–50: I suggest including the recent study by Prytherch and Yelland (2021), which is a dedicated investigation of the influence of sea ice on CO2 exchange: https://doi.org/10.1029/2020GB006633. While it is cited in line 89, it appears to be missing from the bibliography.

**Removed:**

However, both of these examples derived their estimates based on very few actual measurements of CO2 exchange in these challenging environments.

**Added:**

More recently, Prytherch and Yelland (2021) used eddy covariance measurements near a central Arctic Ocean lead to develop a lead-specific gas transfer velocity parameterization during the

summer to fall transition period. Additionally, summertime ship-based Arctic eddy covariance measurements by Dong et al. (2021) showed that surface stratification of fresher, cooler melt water resulted in lower surface  $pCO_{2w}$  compared to 6-m deep  $pCO_{2w}$ , with resulting implications for estimating carbon budgets of polar oceans. While these efforts identify specific processes, more measurements are required to quantify additional gas exchange processes over the annual cycle, as well as validate previous findings.

**Added the following to the bibliography:**

Prytherch, J., and Yelland, M. J.: Wind, Convection and Fetch Dependence of Gas Transfer Velocity in an Arctic Sea-Ice Lead Determined From Eddy Covariance CO2 Flux Measurements, Global Biogeochem. Cycles, 35, https://doi.org/10.1029/2020GB006633, 2021.

Line 125: It appears that a LI-COR 7500 sensor is installed on the tower, but LICOR7200 does not appear. I see in you 2018 paper, the LI-COR 7500 was used to measure water vapor and CO2 was measured by LICOR7200? Could you clarify here?

If you're referring to Figure 2 in Butterworth and Else (2018), yes, the LI-7500 appears more prominently in the photograph. But the LI-7200 is also in that figure in Panel A, just a few feet down from the anemometer mounted inside the frame of the tower. Both were operational throughout the measurement campaign presented in the current manuscript. The reason for using the 7500 for HL was so that tube attenuation would not alter the measurement (as H2O is sticky when in contact with tube walls). The 7200 was for CO2 flux because we needed to dry the airstream prior to measuring the CO2 mixing ratio in order to avoid contamination of the flux measurement due to the overlapping absorption with water vapor.

Line 180: It would be helpful to provide more explanation of the Kice term, specifically, how it was derived or constrained.

We modified the paragraph on Kice to include more information.

**Changed from the original:**

In the laboratory tank sea ice study of Kotovitch et al. (2016) measurements of  $F_{CO2}$ ,  $pCO_{2air}$ , and  $pCO_{2ice}$  were used to determine  $K_{ice}$  – a parameter that encapsulates both the gas transfer velocity and solubility of  $CO_2$  in ice. Here, we estimate  $pCO_{2ice}$  during periods of full ice cover by setting our measured  $F_{CO2}$  equal to the equation

$$F_{CO2} = K_{ice} \left[ pCO_{2ice} - pCO_{2air} \right], \tag{2}$$

where  $K_{\text{ice}}$  was the gas transfer velocity for ice growth and decay (2.5 and 0.4 mol m-2 d-1 atm-1 respectively) found by Kotovitch et al. (2016).

**To:**

In the laboratory study of Kotovitch et al. (2016),  $F_{CO2}$  was measured in a tank over periods of forming, thickening, and melting sea ice. Supporting measurements of  $pCO_{2air}$  and  $pCO_{2ice}$  enabled the derivation of a gas transfer coefficient ( $K_{ice}$ ) using the following bulk formula:

$$F_{CO2} = K_{ice} \left[ pCO_{2ice} - pCO_{2air} \right]. \tag{2}$$

The  $K_{ice}$  parameter encapsulated both the gas transfer velocity and solubility of CO2 in ice. This was done to avoid estimating solubility using seawater-based functions of temperature and salinity outside the range for values for which they were designed.  $K_{ice}$  during periods of ice growth was 2.5 mol m-2 d-1 atm-1, while for periods of ice decay it was 0.4 mol m-2 d-1 atm-1 (Kotovitch et al. 2016).

Because we did not collect in situ  $pCO_{2ice}$  measurements we could not use Eq. (2) to calculate  $K_{ice}$  for independent verification. Instead, we estimated  $pCO_{2ice}$  during periods of full ice cover using Eq. (2) with measured  $F_{CO2}$  and  $pCO_{2air}$  and the  $K_{ice}$  values for ice growth and decay found by Kotovitch et al. (2016). Comparisons of estimated  $pCO_{2ice}$  to previous in situ measurements were used to determine if the laboratory-derived  $K_{ice}$  values were applicable in field conditions.

Figure 3: Could you include information about wind direction? It seems that some flux data may be missing due to winds coming from the direction of the island?

We added 2 new panels (third row) to show wind direction, as well as the range (shaded area) in which flux measurements are discarded due to winds from aft.

Figure 4: The high-frequency time series is 6-hour averaged. Readers may be interested in the extent to which the observed variability is influenced by EC uncertainty. Could you provide at least a simple estimate of the uncertainty magnitude, include that value in the figure caption, and briefly discuss it in the main text?

**Added the following text:**

As the product of measurements from different instruments, the accuracy of the  $F_{\rm CO2}$  measurement is challenging to quantify without an independent validation, which was not performed. The LI-7200 has a measurement accuracy of  $\pm 1\%$  with an RMS noise of 0.11 ppm at 10 Hz, while the vertical wind speed of the CSAT3 is accurate within  $\pm 0.04$  m s-1 with an RMS noise of 0.0005 m s-1. While the noise can occasionally be larger than the true environmental fluctuations, it has been found to minimally influence the calculated  $F_{\rm CO2}$  because the noise from the separate instruments is uncorrelated and therefore filtered out by the flux calculation (Miller et al., 2010).

An investigation of  $F_{CO2}$  measurement uncertainties from ships indicated a detection limit for a dried, closed-path eddy covariance system of roughly  $|\Delta p CO_2| > 35$  µatm for the mean wind speed observed in this study (Blomquist et al., 2014). The  $\Delta p CO_2$  in the region often exceeds this value (Duke et al., 2021; Sims et al., 2023). Additionally, we expect some reduction in the

detection limit (i.e., increased sensitivity) for this study compared to ship-based studies, because the measurements were from a stationary tower. Therefore, the observations avoid some common sources of uncertainty experienced from moving platforms, such as the needed for a complex wind vector motion correction and tilt effects that degrade the performance of the LI-7200 (Miller et al., 2010; Vandemark et al., 2023).

While we cannot perform a direct assessment of  $F_{\rm CO2}$  uncertainty, we can estimate the order of magnitude of the uncertainty by assessing the variation in  $F_{\rm CO2}$  measurements during periods expected to have stable fluxes. Here we do that by calculating the standard deviation for 6-hour intervals during periods of full ice cover, when diurnal variations in  $F_{\rm CO2}$  were expected to be minimal. The standard deviation across these winter periods had a mean of  $\pm 1.02$  mmol m-2 d-1 and a median of  $\pm 0.75$  mmol m-2 d-1. Spring and summer seasons were excluded from the estimate because standard deviation measured during those periods was expected to be a combination of measurement uncertainty and actual diurnal  $F_{\rm CO2}$  trends.

**Added the following text to the Fig. 4 caption:**

Uncertainty in the  $F_{CO2}$  measurement was quantified by calculating the standard deviation from each 6-hour average (comprised of eighteen 20-minute flux intervals) during periods of full ice cover, when diurnal  $F_{CO2}$  variations were minimal. The standard deviations across these winter periods had a mean of  $\pm 1.02$  mmol m-2 d-1 and a median of  $\pm 0.75$  mmol m-2 d-1.

**Added the following reference to the bibliography:**

Vandemark, D., Emond, M., Miller, S. D., Shellito, S., Bogoev, I., and Covert, J. M.: A CO2 and H2O Gas Analyzer with Reduced Error due to Platform Motion. J Atmos Ocean Technol, 40, 845–854. doi: 10.1175/JTECH-D-22-0131.1, 2023

Figure 5: I agree with the other reviewer that the possibility of pCO2ice being negative should be explained. I suspect the derived values may be sensitive to the estimation of Kice. While some discussion is included later in the manuscript, it would be helpful to provide an earlier explanation, perhaps around line 185.

**Added the following in the Spring Discussion section:**

However, it is worth noting that  $p\text{CO}_{2\text{ice}}$  (Fig. 5c) occasionally dropped below zero, which is a physically impossible value. Such instances may indicate that the  $K_{\text{ice}}$  value used to calculate  $p\text{CO}_{2\text{ice}}$  was too small. Because  $K_{\text{ice}}$  combines both gas transfer velocity and solubility, inaccuracies in either term could be responsible. However, it is also possible that the negative values of  $p\text{CO}_{2\text{ice}}$  are simply due to the random error inherent in eddy covariance systems. Because random error can cause both positive and negative deviations in measured flux, these data points were retained to avoid biasing the average.

Additional information about the estimation of Kice was provided in Section 2.3.2. Those changes are shown above in response to the comment about Line 180 above.

Line 275: You mention that camera images were collected, but none are shown in the paper, which is a shame. Would it be possible to include several representative images from different stages of the observation period? These could be placed alongside Table 1 or included in the supplementary material.

Figure 2 was already included. It shows exactly what you are requesting.

Line 333: For your reference, we have conducted a related study using eddy covariance and pCO2w measurements in an ice melt region, which indicates substantial CO2 uptake: Dong et al. (2021), Geophysical Research Letters, https://doi.org/10.1029/2021GL095266.

Thank you for pointing us to this manuscript. It is relevant to our Introduction. We have added the following text:

Additionally, summertime ship-based Arctic eddy covariance measurements by Dong et al. (2021) showed that surface stratification of fresher, cooler melt water resulted in lower surface  $pCO_{2w}$  compared to 6-m deep  $pCO_{2w}$ , with resulting implications for estimating carbon budgets of polar oceans.

Added the following to the bibliography:

Dong, Y., Yang, M., Bakker, D. C. E., Liss, P. S., Kitidis, V., Brown, I., Chierici, M., Fransson, A., and Bell, T. G.: Near-Surface Stratification Due to Ice Melt Biases Arctic Air-Sea CO2 Flux Estimates, Geophys. Res. Lett., 48, https://doi.org/10.1029/2021GL095266, 2021.

Line 370: The first sentence in this paragraph reads awkwardly to me. Please consider rephrasing for improved clarity and flow.

Rephrased from the original:

Figure 8 shows the seasonal temperature dependence of FCO2 on water temperature.

To:

Figure 8 shows the FCO2 dependence on water temperature as it varies across seasons.

Line 533: The square brackets around the reference should be removed to maintain consistency with the formatting style.

Most style guides recommend using square brackets within parentheses when nesting parenthetical information. There was no reference to the subject in the Copernicus Style Guide. Therefore, we would prefer to leave them as they are, to differentiate the levels of information being presented within the parentheses.

Line 569: The abbreviation "EC" appears here without prior definition.

Changed "EC" to "eddy covariance"

Conclusions: The comparison with Sims et al. (2023) is valuable, but might be more impactful if introduced earlier in the discussion section. As currently presented, it reads more like a discussion point than a concluding remark.

Moved Sims et al. (2023) discussion from Conclusions to Process Summary section, and added additional text. It now reads:

"The direction of fluxes that we measured across the annual cycle were in general agreement with  $\Delta p \text{CO}_2$  gradients measured by Sims et al. (2023) within a ~100 km radius of the flux station. Sims et al. (2023) did note substantial spatial variability, which makes it difficult to confidently extrapolate the net annual flux over a larger area. However, an estimate of k calculated using tower  $F_{\text{CO}_2}$  and ship-based  $p \text{CO}_{2w}$  measurements of Sims et al. (2023) during temporally-aligned courses past the island showed good agreement with existing open-water k parameterizations, providing evidence the capability of the tower-based  $F_{\text{CO}_2}$  for estimating  $p \text{CO}_{2w}$  (Butterworth and Else, 2018)."

Final suggestion: It may strengthen the conclusions if you emphasize that concurrent measurements of pCO2w would provide more robust support for some of the interpretations presented in this study.

**Added:**

Future research from this site may be able to highlight the magnitude of individual processes with greater precision. Due to its relevance to the  $F_{\rm CO2}$  cycle, direct measurements of  $p{\rm CO}_{\rm 2w}$  were collected at the site during subsequent years. These were made possible by the installation of a mobile power station/research lab (with sleeping quarters), installed on the island in 2018. These measurements will be incorporated into future research investigating  ${\rm CO}_2$  gas transfer velocity continuously through the annual cycle.

**-Yuanxu Dong**

**Added to the Acknowledgements:**

We would also like to thank Yuanxu Dong and one anonymous reviewer for their constructive reviews.